

# High-dimensional normalized data profiles for testing derivative-free optimization algorithms

Hassan Musafer[1], Emre Tokgoz[2] and Ausif Mahmood[1]

[1] School of Computer Science and Engineering, University of Bridgeport, Bridgeport, CT, United States of America
[2] School of Engineering, Quinnipiac University, Hamden, CT, United States of America

## ABSTRACT

This article provides a new tool for examining the efficiency and robustness of derivative-free optimization algorithms based on high-dimensional normalized data profiles that test a variety of performance metrics. Unlike the traditional data profiles that examine a single dimension, the proposed data profiles require several dimensions in order to analyze the relative performance of different optimization solutions. To design a use case, we utilize five sequences (solvers) of trigonometric simplex designs that extract different features of non-isometric reflections, as an example to show how various metrics (dimensions) are essential to provide a comprehensive evaluation about a particular solver relative to others. In addition, each designed sequence can rotate the starting simplex through an angle to designate the direction of the simplex. This type of features extraction is applied to each sequence of the triangular simplexes to determine a global minimum for a mathematical problem. To allocate an optimal sequence of trigonometric simplex designs, a linear model is used with the proposed data profiles to examine the convergence rate of the five simplexes. Furthermore, we compare the proposed five simplexes to an optimized version of the Nelder–Mead algorithm known as the Genetic Nelder–Mead algorithm. The experimental results demonstrate that the proposed data profiles lead to a better examination of the reliability and robustness for the considered solvers from a more comprehensive perspective than the existing data profiles. Finally, the high-dimensional data profiles reveal that the proposed solvers outperform the genetic solvers for all accuracy tests.

# INTRODUCTION

The growing success in developing derivative-free optimization (DFO) algorithms and applications has also motivated researchers over the past decades to provide new tools for DFO performance analysis. The purpose of these tools is that when a new DFO algorithm/solver is presented into the optimization literature, it is expected to comprehensively evaluate its performance against other similar algorithms. This is required to secure a fair comparison as a basis to evaluate the relative performance of different solvers. In addition, the developed measurement scheme for comparison between

Corresponding author
Hassan Musafer,
hmusafer@bridgeport.edu

similar algorithms needs to examine the level of complexity in the algorithm design, and computes the computational budget required by the algorithm compared to others (*Vince & Earnshaw, 2012*).

We are motivated by the observation that most algorithm developers are interested in testing one performance measure (one dimension). For example, some data profiles are designed to provide for users with information about the percentage of solved problems as a function of simplex gradient estimates (*More & Wild, 2009*). However, if the evaluation is expensive, one dimension may not provide useful information to capture how reliable a solver performs relative to the other solvers, as we will demonstrate later. In order to provide a comprehensive evaluation for the relative performance of multiple solvers, we introduce a collection of performance metrics to evaluate new algorithms and improve the existing data profiles. Numerical results indicate that the proposed high-dimensional data profiles are more compact and effective in allocating a computational budget for different levels of accuracy.

The focus of our work is exclusively on minimization problems. Such problems arise naturally in almost every branch of modern science and engineering. For example, pediatric cardiologists seek to delay the next operation as much as possible to identify the best shape of a surgical graft (*Audet & Hare, 2017*). In this particular example, a number of variables can affect the objective function to treat and manage heart problems in children. Some are structural differences they are born with, such as holes between chambers of the heart, valve problems, and abnormal blood vessels. Others involve abnormal heart rhythms caused by the electrical system that controls the heart beat. Technically, we can write the minimum function value of $f$ over the constraint set $\Omega$ in the form.

$$\min_x\{f(x) : x \in \Omega\}. \tag{1}$$

Note that, the minimum function value could be:

i. $-\infty$: such as

$$\min_x\{x_1 : x \in \mathbb{R}^3\}.$$

ii. A well-defined real number: such as

$$\min_x\{\|x\|^2 : x \in \mathbb{R}^2, \ x_1 \in [-1, 2], \ x_2 \in [0, 3]\}.$$

However, there are other equivalent forms. Suppose that a researcher is interested in obtaining an estimate of the point or set of points that determine the minimum function value $z$ (*Audet & Hare, 2017*). We might instead seek the argument of the minimum:

$$\text{Argmin}_x\{f(x) : x \in \Omega\} := \{x \in \Omega : f(x) = z\} \tag{2}$$

In particular, the argmin set can be:

i. A singleton: such as

$$\text{argmin}_x\{\|x\|^2 : x \in \mathbb{R}^2, \ x_1 \in [-1, 2], \ x_2 \in [0, 3]\} = \{[0, 0]^T\}.$$

ii. A set of points: such as

$$\underset{x}{argmin}\{sin(x) : x \in \mathbb{R}, \ x_1 \in [0,7]\} = \{0, \pi, 2\pi\}.$$

One of the most common examples of derivative-free optimization algorithms is the Nelder Mead simplex gradient algorithm (1965) (NMa), which is one of the widely used algorithms for minimization problems (*Barton & Ivey Jr, 1996*; *Lewis, Torczon & Trosset, 2000*; *Wright et al., 2010*; *Lagarias et al., 1998*; *Wouk et al., 1987*). The notion of the NMa is based on creating a geometrical object, called simplex, in the hyperplanes of n-parameters. Then, this simplex performs reflections over the changing solution space of a mathematical problem until the coordinates of the minimum point can be obtained by one of its vertices (*Spendley, Hext & Himsworth, 1962*; *Kolda, Lewis & Torczon, 2003*).

The contribution of the NMa is to incorporate the simplex search with non-isometric reflections, designed to accelerate the search (*Lewis, Torczon & Trosset, 2000*; *Conn, Scheinberg & Vicente, 2009*; *Han & Neumann, 2006*). It was well-understood that the non-isometric reflections of NMa were designed to deform the simplex in a better way to explore the solution space of mathematical functions (*Lewis, Torczon & Trosset, 2000*; *Baudin, 2009*). Nevertheless, when the number of parameters under investigation increases, the simplex becomes increasingly distorted with each iteration, generating different geometrical formations that are less effective than the initial simplex design (*Baudin, 2009*; *Torczon, 1989*). In addition, *McKinnon (1998)* analyzed the original NMa for strictly convex functions with up to three continuous derivatives. In all the objective functions, the NMa causes the sequence of the generated simplexes to converge to a non-stationary point. The NMa repeats inside construction steps with the best vertex remaining fixed; until the diameter of the simplex approximately shrinks to 0.

A recent contribution to the NMa is the Genetic Nelder Mead algorithm (GNMa) that hybridizes NMa with genetic programming (*Fajfar, Puhan & Burmen, 2017*). The GNMa evolves improved vertices using cross-over and mutation operations to initialize new simplex designs better than the traditional method for initializing a simplex. Thus, the new algorithm generates many population-based simplexes with different shapes and keeps the best designs that have better features to locate an optimal solution. The authors have only one issue with the original NMa, which is the reduction step. They claim that this operation is inconsistent because it does not return a single vertex. They suggested that the reduction step should include exclusively the worst vertex and that, basically, the inner contraction can perform the job. The new implementation of the GNMa performs four operations: reflection, expansion, inner contraction, and outer contraction. In addition to the three basic vertices of the original NM, the authors add one more vertex, defined as the second best. The new vertex joins the other basic vertices to constitute a centroid different than the one that was defined by *Nelder & Mead (1965)*. The GNMa forms the next simplex by reflecting the vertex that is associated with the highest value of the cost function (CF), in the hyperplane spread over the remaining vertices.

The main aim of this research is to improve the existing data profiles by adding a variety of metric measures for testing DFO algorithms. In addition, we propose five sequences of trigonometric simplex designs that work separately to optimize the individual components

of mathematical functions. To allocate the optimal sequence of the triangular simplex designs, a linear model with a window of 10 samples is proposed for evaluating the multiple simplexes (solvers) in the neighborhood of the minimum. The rest of this article is organized as follows: The next section presents the theory of the sequential design of the trigonometric Nelder–Mead algorithm, and demonstrates a compact mathematical way of implementing the algorithm based on vector theory. 'Multidirectional trigonometric Nelder Mead' describes the importance of the initial simplex design, and presents the multidirectional trigonometric Nelder Mead algorithm (MTNMa). 'Computational experiments' presents data profiles and statistical experiments to compare the reliability and robustness of the MTNMa to that of the GNMa (*Fajfar, Puhan & Burmen, 2017*) on standard test functions (*More, Garbow & Hillstrom, 1981*). Finally, the conclusions are provided in 'Conclusion'.

## HASSAN NELDER MEAD ALGORITHM

We present in this section the theory of the Hassan Nelder Mead algorithm (HNMa) (*Musafer & Mahmood, 2018*; *Musafer et al., 2020*), and describe the importance of the dynamic properties of the algorithm that make it appropriate solution for unconstrained optimization problems. The sequential trigonometric simplex design of the HNMa allows components of the reflected vertex to adapt to different operations; by breaking down the complex structure of the simplex into multiple triangular simplexes. This is different from the original NMa that forces all components of the simplex to execute a single operation such as expansion. When different reflections characterize the next simplex, the HNMa performs similar reflections to that of the original simplex of the NMa and others with different orientations determined by the collection of non-isometric features. As a consequence, the generated sequence of triangular simplexes is guaranteed to search a higher proportion of the solution space and performs better than the original simplex of the NMa (*Nelder & Mead, 1965*).

We now present a mathematical way of analyzing the HNMa using vector theory, and explain why the original NMa fails in some instances to find a minimal point or converges to a non-stationary point. For example, suppose that we want to determine the minimum of a function $f$. The function $f(x, y)$ is calculated at vertices that are subsequently arranged in ascending order with respect to the CF values, such that: $A(x_1, y_1) < B(x_2, y_2) < C(x_3, y_3) < Th(x_4, y_4)$, where $A, B$, and $C$ are the vertices of the triangular simplex with respect to the lowest, 2nd lowest, and 2nd highest CF values, and $Th$ is a threshold that has the highest CF value. The need for the $Th$ is when the HNMa performs a reflection in a dimension (such as x) of the solution space of the $f(x, y)$, the computed value in that dimension replaces the axial value of $(x)$ in the $Th$. Once the axial values of the $Th(x, y)$ are updated, and the position of the $Th$ leads to lower CF value than the previous CF value of the $Th$, then the HNMa moves to upgrade the $Th$. After upgrading the $Th$ point with a variety of non-isometric reflections, the HNMa examines the $Th$ to validate if the resulted $Th$ has a lower CF value than $C$ to be replaced with $C$ or the HNMa needs to upgrade the $Th$ only. This technique of exploring the neighborhood

of a minimum is to search the optimal pattern that can be followed and result in a better approach to the minimum.

To construct a triangular simplex of the HNMa, we need to find three key midpoints: $H, I$, and $G$, as seen in Fig. 1 part–I. They are found by calculating the average coordinates of the connected line segments $(A$ and $B)$, $(A$ and $C)$, and $(C$ and $H)$ respectively. Hence to simplify the problem, our analysis depends on the combinations of $x$–components and $y$–components (if there are more components, then we append the following equations with the additional axial components), to satisfy,

$$H(x_5, y_5) = \frac{A+B}{2} = \left( x_5 = \frac{x_1 + x_2}{2}, y_5 = \frac{y_1 + y_2}{2} \right) \tag{3}$$

$$I(x_6, y_6) = \frac{A+C}{2} = \left( x_6 = \frac{x_1 + x_3}{2}, y_6 = \frac{y_1 + y_3}{2} \right) \tag{4}$$

$$G(x_7, y_7) = \frac{H+C}{2} = \left( x_7 = \frac{x_5 + x_3}{2}, y_7 = \frac{y_5 + y_3}{2} \right) \tag{5}$$

Note that to find the reflected point $D$, we add the vectors $H$ and $d$, as shown in Fig. 1 part-I, where $d$ is the vector that can be represented by subtracting any of the vectors $(H$ and $C)$, $(D$ and $H)$, $(E$ and $D)$ or $(F$ and $G)$. The coordinates of $D$ are obtained by adding the vectors $(H)$ and $(d)$. The vector formula is given below.

$$D = H + d = H + (H - C) = 2H - C = (2x_5 - x_3, 2y_5 - y_3) \tag{6}$$

A similar process could be used to find the coordinates of $E$ and $F$. The formulas are stated below.

$$E = H + 2d = H + 2(H - C) = 3H - 2C = (3x_5 - 2x_3, 3y_5 - 2y_3) \tag{7}$$

$$F = H + d_1 = H + (H - G) = 2H - G = (2x_5 - x_7, 2y_5 - y_7) \tag{8}$$

where $d_1$ can be found by subtracting any of the vectors $(G$ and $C)$, $(H$ and $G)$, $(F$ and $H)$ or $(D$ and $F)$. Hence, the HNMa does not have a shrinkage step; instead, two operations are added to the algorithm: shrink from worse to best $I(x_6, y_6)$ and shrink from good to best $H(x_5, y_5)$. The basic six reflections of the HNMa are shown in Fig. 1 part-II.

It is noteworthy to mention that a combination of $x$–components of the HNMa results in the extraction one of the six non-isometric reflections. Now, if we consider two combinations (such as x and y) or more, then the simplex as in the case of the HNMa performs two reflections or more. Thus, the multiple components of the triangular simplex adapt to extract various non-isometric features of the HNMa. Therefore, the optimization solution of the HNMa reflects the opposite side of the simplex through the worse vertex and leads to the implementation of reflections determined by the collection of extracted

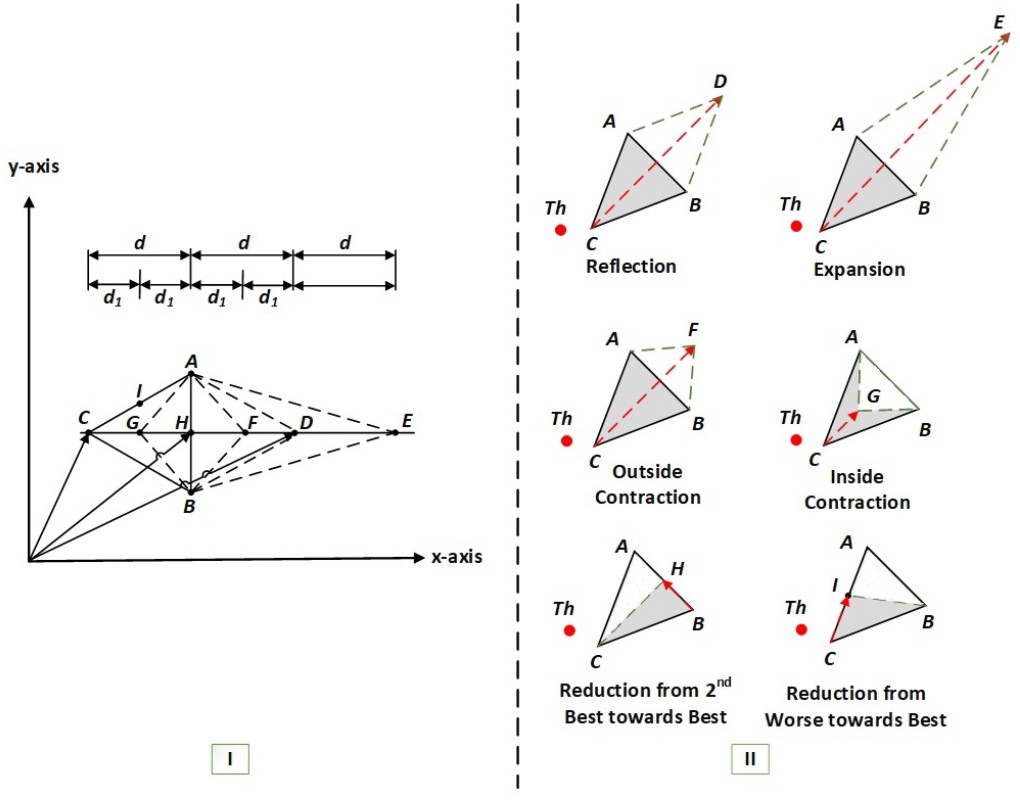

**Figure 1** I–The geometrical analysis of an HNMa based on vector theory (*Musafer & Mahmood, 2018*), II–the basic six operations of an HNMa (*Musafer et al., 2020*).

features. For example, suppose we need to find the minimum of a function $f(x, y)$. A solution of the NMa may come out to be reflection in $x$ and $y$ directions, whereas a solution of the HNMa may come out to be reflection in $x$ but expansion in $y$. It can be a combination of any two reflections of the HNMa. In fact, the HNMa is designed to deform its simplex in a way that is more adaptive to tackle the optimization problems than the original simplex of the NMa. The triangular simplexes of the HNMa extract different non-isometric reflections from different dimensions; therefore, the reflected vertex is rotated through an angle to produce simplexes that lead to faster convergence rates than the original triangular simplex of the NMa.

## MULTIDIRECTIONAL TRIGONOMETRIC NELDER MEAD

The Nelder and Mead algorithm is particularly sensitive to the position of the initial simplex design, where the variable-shape simplex is modified at each iteration using one of four linear operations: reflection, expansion, contraction, and shrinkage. The geometrical shape of the simplex subsequently becomes distorted as the algorithm moves towards a minimal point by generating different geometrical configurations that are less effective than the initial simplex design. To address this need, one of the preferred designs is to build the initial simplex with equal length edges (*Martins & Lambe, 2013*). In this way, the

unit simplex of dimension $n$ is shifted from the origin to the initial guess. Suppose that the length of all sides of the simplex is required to be $l$. The given starting point $x_0$ of dimension $n$, is the initial vertex $v_1 = x_0$. We define the parameters $a, b > 0$ as follows:

$$b = \frac{l}{n\sqrt{2}}(\sqrt{n+1} - 1) \tag{9}$$

$$a = b + \frac{l}{\sqrt{2}} \tag{10}$$

The remaining vertices are computed by adding a vector to $x_0$; whose components are all $(b)$ values except for the $j$th component that is assigned to $(a)$, where $j = 1, 2, \ldots, n$, and $i = 2, 3, \ldots, n+1$, as follows.

$$v_{i,j} = \begin{cases} x_{0,j} + a & \text{if } j = i - 1 \\ x_{0,j} + b & \text{if } j \neq i - 1 \end{cases} \tag{11}$$

The risk is that if the coordinate's direction of the constructed initial simplex is perpendicular to the direction towards the minimal point, then the algorithm performs a large number of reflections or converges to a non-stationary point (*McKinnon, 1998*). The practical problem of designing such an initial simplex lies in two parameters: the initial length and the orientation of the simplex. As a result, this simplex is not very effective, especially for problems that involve more than 10 variables (*Martins & Lambe, 2013*).

Alternatively, the most popular way of initializing a simplex is Pfefferś method, which is due to L. Pfeffer at Stanford (*Baudin, 2009*). The method is heuristic and builds the initial simplex with respect to the characteristics of the starting point $x_0$. The method adjusts the orientation and size of a simplex by modifying the values of usual delta $(\delta_u)$ and zero term delta $(\delta_z)$ elements. Pfeffer's method is presented in *Fan (2002)* and used in the "fminsearch" function from the "neldermead package" (*Bihorel, Baudin & Bihorel, 2018*). To build a simplex as suggested by L.Pfeffer, the initial vertex is set to $v_1 = x_0$, and the remaining vertices are obtained as follows,

$$v_{i,j} = \begin{cases} x_{0,j} + \delta_u * x_{0,j} * i & \text{if } j = i - 1 \text{ and } x_{0,j} \neq 0 \\ \delta_z & \text{if } j = i - 1 \text{ and } x_{0,j} = 0 \\ x_{0,j} & \text{if } j \neq i - 1 \end{cases} \tag{12}$$

The positive constant coefficients of $\delta_z$ and $\delta_u$ are selected to scale the initial simplex with the characteristic length and orientation of the $x_0$. The vertices are $i = 2, 3, \ldots, n+1$, and the parameters of the vertices are $j = 1, 2, \ldots, n$. If the constructed simplex is flat or is not in the same direction as an optimal solution, then this initial simplex may fail to drive the process towards an optimum or require to perform a large number of simplex evaluations. Therefore, the selection of a good starting vertex can greatly improve the performance of the NMa.

On the contrary, our strategy is to allow the components of the reflected vertex to perform different reflections of the HNMa. This means that each triangular simplex performs one type of reflections regardless of the reflections implemented by the other

triangular simplexes. Therefore, we form the initial triangular simplexes with similar scaling characteristics and with respect to the features of the starting point. In addition, we reinforce the traditional simplex design of the HNMa with four additional simplex designs. The five simplexes are multidirectional and designed to explore the solution space and allocate distinct non-isometric reflections and phase rotations for approaching a global minimal value.

To initialize a simplex of the HNMa (*Musafer & Mahmood, 2018*), Eq. (12) is modified to be consistent with the new requirements of the HNMa, as follows.

$$v_{i,j} \, (Solver\,1) = \begin{cases} x_{0,j} + \delta_u * x_{0,j} * i & \text{if } x_{0,j} \neq 0 \\ x_{0,j} + \delta_z * i & \text{if } x_{0,j} = 0 \end{cases} \tag{13}$$

According to *Gao & Han (2012)*, the default parameter values for $\delta_u$ and $\delta_z$ are 0.05 and 0.00025 respectively. The indices of the HNMa simplex used are $i = 2, \ldots, 5$, and $j = 1, 2, \ldots, n$ (*Musafer & Mahmood, 2018*).

In this test, we are more interested in launching multiple sequences of trigonometric simplex designs that extract various non-isometric reflections and perform different phase rotations. Each sequence is designed to rotate the starting simplex through an angle that designates the direction of the simplex. The proposed MTNMa enhances the standard HNMa of constructing a simplex by adding other designs for high performing optimization algorithm. We will demonstrate how solvers of the MTNMa extract different features of non-isometric reflections and converge to a minimum with a smaller computational budget as compared to the previously discussed methods of simplex designs. Key to this outcome is the mathematical model of the MTNMa designed to determine the optimal features of non-isometric reflections that result in better approximate solutions as compared to optimized versions of simplex designs.

One of the potential simplex designs is to multiply the odd-indexed variables of odd-indexed vertices by (-1); the values of $\delta_z$ and $\delta_u$ are modified to perform a reflection in the y-components of the triangular simplexes of Solver1. The formula is as follows:

$$v_{i,j} \, (Solver\,2) = \begin{cases} x_{0,j} + (-1)^j * \delta_u * x_{0,j} * i * \bmod\left(\frac{i+j}{2}\right) & \text{if } x_{0,j} \neq 0 \text{ and } \bmod\left(\frac{i+j}{2}\right) = 1 \\ x_{0,j} + (-1)^j * \delta_z * i * \bmod\left(\frac{i+j}{2}\right) & \text{if } x_{0,j} = 0 \text{ and } \bmod\left(\frac{i+j}{2}\right) = 1 \end{cases} \tag{14}$$

Similarly, we can obtain a mirror image of the above formula if we apply the transformation on the even components of $x_0$ to generate new vertices. Solver4 performs a reflection in the x-components of the triangular simplexes of Solver1. The corresponding equation is as follows.

$$v_{i,j} \, (Solver\,4) = \begin{cases} x_{0,j} + (-1)^{j+1} * \delta_u * x_{0,j} * i * \bmod\left(\frac{i+j}{2}\right) & \text{if } x_{0,j} \neq 0 \text{ and } \bmod\left(\frac{i+j}{2}\right) = 0 \\ x_{0,j} + (-1)^{j+1} * \delta_z * i * \bmod\left(\frac{i+j}{2}\right) & \text{if } x_{0,j} = 0 \text{ and } \bmod\left(\frac{i+j}{2}\right) = 0 \end{cases} \tag{15}$$

A different way to create a simplex design that differs from Solver1, Solver2, and Solver4, is to push some or all the points of the Solver1 towards the negative (x and y) axes to constitute Solver3 or towards the positive axes to constitute Solver5. Hence,

Solver3 rotates the triangular simplexes of Solver1 by 180 degrees about the origin, which is obtained by multiplying the odd and even components of (x and y) by ($-1$). Similarly, Solver5 is designed to adjust the simplexes of Solver1 to perform a reflection in $x$-axis, $y$-axis, or origin, which is obtained by taking the absolute value of the triangular simplexes of Solver1. The corresponding formulas are as follows:

$$v_{i,j}\ (Solver3) = \begin{cases} x_{0,j} - \delta_u * x_{0,j} * i & \text{if } x_{0,j} \neq 0 \\ x_{0,j} - \delta_z * i & \text{if } x_{0,j} = 0 \end{cases} \quad (16)$$

$$v_{i,j}\ (Solver5) = \begin{cases} x_{0,j} + \delta_u * \|x_{0,j}\| * i & \text{if } x_{0,j} \neq 0 \\ x_{0,j} + \delta_z * i & \text{if } x_{0,j} = 0 \end{cases} \quad (17)$$

To monitor and evaluate a sequence of trigonometric simplex design, we need to know two points that the simplex has passed through as well as the slope with respect to their CF values. Therefore, a window of size 10 points is used to examine the simplex performance. The window size is derived from our practical experience. One of the proposed solvers manages to locate the exact minimum for (Jennrich-Sampson) function within 22 simplex evaluations. Based on the evaluation of the direction vector, the simplex is either allowed to continue exploring the solution space or aborted. Consider a simplex that has passed through a window of 10-points, we need to know the first point $P_1(x_1, y_1)$ and the last point $P_{10}(x_{10}, y_{10})$ of the window as well as the direction of the simplex. We can write this as a line in the parametric form by using vector notation.

$$\langle x, y \rangle = \langle x_1, y_1 \rangle + t \langle m_x, m_y \rangle \quad (18)$$

For the particular case, we can select $\langle x_1, y_1 \rangle = P_1 \langle x_1, y_1 \rangle$, so the direction vector is found as follows:

$$\langle m_x, m_y \rangle = P_{10} \langle x_{10}, y_{10} \rangle - P_1 \langle x_1, y_1 \rangle \quad (19)$$

If the coordinates of the direction vector equal zero, this indicates that all best points that the simplex (solver) has passed through had equal coordinates, then the simplex is aborted unless it satisfies a convergence test based on the resolution of the simulator. The observing process continues for all the sequences of triangular simplexes on the coordinate plane until the coordinates of the minimal point are found by one of the simplex designs that needs less computational budget than the others. Another advantage of using Eq. (19), when combined with data profiles later to evaluate several solvers, this formula can be used as a criteria to stop a solver that cannot satisfy the convergence test within the given computational budget.

## COMPUTATIONAL EXPERIMENTS

In this section, we present the test procedures that provide a comprehensive performance evaluation of the proposed algorithm. We follow two stages to carry out the experiments. In the first stage, we define the metrics that differentiate between the considered algorithms,

which are summarized as follows: the accuracy of the algorithm compared to the actual minima, the wall-time to convergence (in seconds), the number of function evaluations, the number of simplex evaluations, and identification of the best sequence of trigonometric simplex designs. In addition, we adopt the guidelines designed by *More, Garbow & Hillstrom (1981)*, to evaluate the reliability and robustness of unconstrained optimization software. These guidelines utilize a set of functions exposed to an optimization algorithm to observe weather the algorithm is tuned to particular functions that belong to one type of optimization class or not. For this purpose, *More, Garbow & Hillstrom (1981)* introduced a large collection of different optimization functions for evaluating the reliability and robustness of unconstrained optimization software. The features of the test functions cover three classes: nonlinear least squares, unconstrained minimization, and systems of nonlinear equations.

The second stage involves normalized data profiles suggested by *More & Wild (2009)* with a convergence test given by the formula (??). The function of data profiles is to provide an accurate view of the relative performance of multiple solvers belonging to different algorithms when there are constraints on the computational budget.

$$f(x_0) - f(x) \geq (1 - \tau)(f(x_0) - f_L) \tag{20}$$

where $x_0$ is the starting point for the solution of a particular problem $p, p \in P$ $(P)$ is a set of benchmark problems), $f_L$ is the smallest CF value obtained for the problem by any solver within a given number of simplex gradient evaluations, and $\tau = 10^{-k}$ is the tolerance with $k \in \{3, 5, 7\}$ for short-term outcomes. These include changes in adaptation, behavior, and skills of derivative-free algorithms that are closely related to examining the efficiency and robustness of optimization solvers at different levels of accuracy.

In this research, however, the MTNMa launches multiple solvers that compute a set of approximate solutions. The definition of the convergence test (??) is independent of determining the different optimization solvers that satisfy a certain accuracy, as in the case of algorithms that generate multiple solvers. This is not realistic, solvers mostly cannot approximate to an optimal solution in a similar number of evaluations, thereby some solvers may push the process faster towards the optima than others. Therefore, we use a linear model that has already been defined as the criteria for stopping the algorithm if one of the solvers satisfies a convergence test within a limited computational budget. Assume that we have a set of optimization solvers $S$ converging to best possible solution $f_L$ obtained by any solver within a given number of simplex evaluations. The convergence test used for measuring several relative distances to optimality can be defined with respect to $s$, $(s \in S)$, we might instead write the convergence test in the following form:

$$f(x_0) - f^s(x) \geq (1 - \tau)(f(x_0) - f_L) \tag{21}$$

The previous work with data profiles has assumed that the number of simplex evaluations (one dimension) is the dominant performance measure for testing how well a solver performs relative to the other solvers (*More & Wild, 2009*; *Audet & Hare, 2017*). However, they did not investigate the performance of derivative free optimization solvers if a variety of metrics were used to evaluate the performance. If the cost unit is evaluated only

using simplex evaluations, then this assumption is unlikely to hold, when the evaluation is expensive, as we will demonstrate later. In this case, we might instead define the performance measures to be the amount of computational time and number of simplex evaluations. Specifically, we define data profiles in terms of a variety of performance metrics, summarized: the amount of computational time $T$, the number of simplex evaluations $W$, the number of function evaluations $Y$, and the number of CPU cores $Z$ required to satisfy the convergence test (??). We thus define the data profile of a solver $s$ by the formula.

$$d^s(T,W,Z) = \frac{1}{\|P\|} size \left\{ p \in P : \frac{t^s(p)}{n_p+1} \leq T, \frac{w^s(p)}{n_p+1} \leq W, \frac{y^s(p)}{n_p+1} \leq Y, \frac{z^s(p)}{n_p+1} \leq Z \right\} \quad (22)$$

where $\|P\|$ denotes the cardinality of $P$, $n_p$ is the number of variables $p \in P$, and $t^s(p), w^s(p), y^s(p)$ *and* $z^s(p)$ are the performance metrics for timing the algorithm, counting number of simplex evaluations, counting number of function evaluations, and counting number of CPU cores respectively.

Altogether, the computational experiments are conducted to evaluate the MTNMa on a computer that has 1.8 GHz core i5 CPU and 4 GB RAM. Finally, C# language is used to implement the MTNMa and the experiments.

## Discussion

The HNMa generates a sequence of triangular simplexes that extract a collection of non-isometric reflections to calculate the next vertex. Each simplex crawls independently to adapt its shape to the solution space of unconstrained optimization problems. Therefore, the convergence speed per simplex varies from one iteration to another. A simplex in some cases explores the neighborhood to update its threshold, but moves only if the threshold is good enough to replace the worst point. However, in other cases the simplex continues to generate different triangular shapes and orientations. Therefore, the generated simplexes of the HNMa extract different features of non-isometric reflections to update the simplexes with optimal triangular shapes and rotations. In this way, the HNMa mimics an amoeba style of maneuvering from one point to another when approaching a target (minimal point). On the contrary, the NMa (*Nelder & Mead, 1965*) forces components of the reflected vertex to follow one of four linear operations (reflection, expansion, contraction, and shrinkage). When the next vertex is characterized by one operation (one type of reflections), some dimensions of the reflected vertex depart for less optimal values. This problem obviously appears in high-dimensional applications. Consequently, the simplex shapes of the NMa becomes less effective in high dimensions and tends to deteriorate rapidly with each iteration. The HNMa (*Musafer & Mahmood, 2018*) has proven to deliver a better performance than the traditional NMa, represented by a famous Matlab function, known as "*fminsearch*".

To promote the traditional simplex design of the HNMa, MTNMa generates five sequences of trigonometric simplex designs. Some points in the initial sequence of triangular simplexes of HNMa (Eq. (13)) are perturbed and used as starting points to launch other simplex designs with different reflections. For example, (Eq. (14)) the triangular simplexes of Solver2 are obtained by reflecting the y-components of the

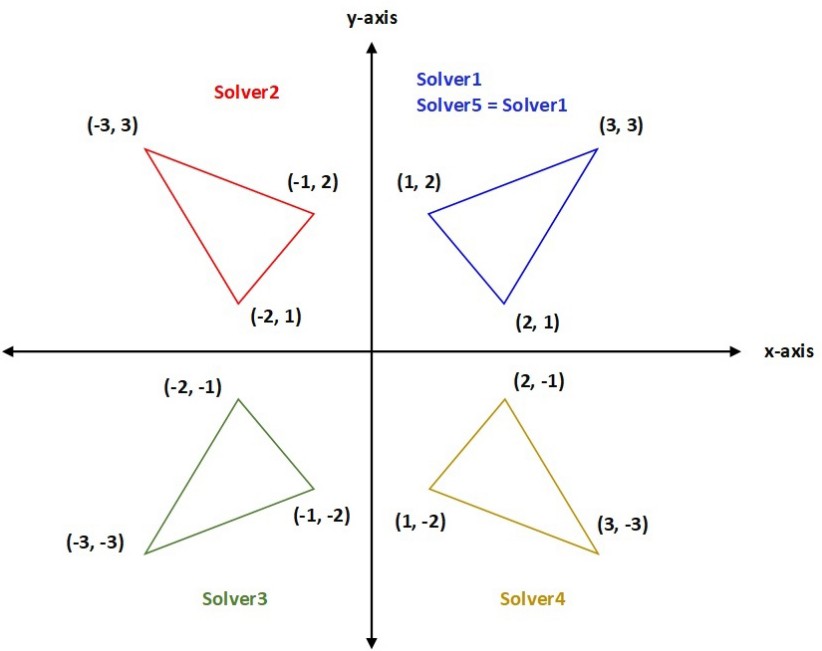

**Figure 2** An example of different formations of Solver1.

triangular simplexes of Solver1, which is performed by multiplying the x-components of Solver1 by (−1). Similarly, (Eq. (16)) the triangular simplexes of Solver1 are rotated 180 degrees to constitute the triangular simplexes of Solver3 (same as reflection in origin), which is obtained by multiplying the (x and y) components of Solver1 by (−1). (Eq. (15)) the triangular simplexes of Solver4 are initialized by reflecting the x-components of the triangular simplexes of Solver1, which is achieved by multiplying the y-components of Solver1 by (−1). Finally, (Eq. (17)) the triangular simplexes of Solver5 are obtained by taking the absolute value of the triangular simplexes of Solver1. Solver5 can generate triangular simplexes by reflection in the x-coordinate, y-coordinate, or origin, or initialize triangular simplexes that are similar to that of the simplexes of Solver1. Figure 2 shows all the transformations on the (x and y) components of the traditional vertices of Solver1 to generate new vertices for Solver2, Solver3, Solver4, and Solver5. We assume that 3 arbitrary vertices of the triangular simplex (Solver1) shown in Fig. 2 have component values (1, 2), (2, 1), and (3, 3).

Numerical experiments in Table 1 are performed to test the efficiency and robustness of the MTNMa. The purpose of the computational study is to show that the definition of normalized data profiles for testing one dimension (such as simplex evaluations) in some cases is not an accurate measure for comparison between similar algorithms. Thus, one dimension may not reflect enough information to examine the efficiency and robustness of DFO solvers when similar algorithms generate multiple solvers and use the normalized data profiles to allocate the computational budget. For this reason, we propose high-dimensional normalized data profiles that serve as an accurate measure when comparing

**Table 1  Summary of experimental results.**

| Test Function ($n$) | GNMa $\frac{(Acc.)(BestSolver)}{(FunctionEv.)}$ | MTNMa $\frac{(Accuracy)(BestSolver)}{(FunctionEv.)(SimplexEv.)(Time)}$ | Actual Minima |
|---|---|---|---|
| Rosenbrock (2) | 0.0 (2) | 0.0 (1) | 0.0 |
| | (1516) | (6963) (799) (0.0312) | |
| Freudenstein-Roth (2) | 48.9842 (1) | 48.9842 (5) | 48.9842 |
| | (425) | (419) (47) (0.0200) | |
| Powell badly scaled (2) | 0.0 (1) | 0.0 (1) | 0.0 |
| | (1957) | (9738) (694) (0.0156) | |
| Brown badly scaled (2) | 0.0 (1) | 0.0 (2, 4, 5) | 0.0 |
| | (1349) | (1449, 1450, 1431) (196) (0.0155) | |
| Beale (2) | 0.0 (1) | 0.0 (2, 4) | 0.0 |
| | (683) | (1935, 2029) (181) (0.0312) | |
| Jennrich-Sampson (2) | 124.362 (1) | 124.362 (5) | 124.362 |
| | (397) | (212) (22) (0.0156) | |
| Helical valley (3) | 0.0 (2) | 0.0 (3) | 0.0 |
| | (7287) | (22278) (1443) (0.1010) | |
| Bard (3) | $8.2148\ldots 10^{-3}$ (2) | $8.2148\ldots 10^{-3}$ (4) | $8.2148\ldots 10^{-3}$ |
| | (1020) | (1065) (72) (0.0156) | |
| Gaussian (3) | $1.1279\ldots 10^{-8}$ (2) | $1.1279\ldots 10^{-8}$ (2, 4) | $1.1279\ldots 10^{-8}$ |
| | (567) | (442, 467) (36) (0.0156) | |
| Meyer (3) | 87.9458 (1) | 87.9483 (1) | 87.9458 |
| | (4511) | (3776182) (357780) (33.0791) | |
| Box 3D (3) | 0.0 (1) | $2.7523\ldots 10^{-29}$ (1) | 0.0 |
| | (2430) | (517602) (51060) (60.7032) | |
| Gulf research (3) | $2.4074\ldots 10^{-35}$ (2) | $1.5242\ldots 10^{-26}$ (4) | 0.0 |
| | (16186) | (252305) (24600) (33.8493) | |
| Powell singular (4) | $1.9509\ldots 10^{-61}$ (1) | 0.0 (2) | 0.0 |
| | (4871) | (56958) (3878) (0.2031) | |
| Wood (4) | 0.0 (3) | $3.9936\ldots 10^{-30}$ (3) | 0.0 |
| | (4648) | (9871) (500) (0.0468) | |
| Kowalik $-$Osborne (4) | $3.0750\ldots 10^{-4}$ (1) | $3.0750\ldots 10^{-4}$ (4) | $3.0750\ldots 10^{-4}$ |
| | (1206) | (6224) (423) (0.0900) | |
| Brown $-$Dennis (4) | 85822.2 (1) | 85822.2 (5) | 85822.2 |
| | (1288) | (1322) (76) (0.0781) | |
| Quadratic (4) | 0.0 (2) | 0.0 (5) | 0.0 |
| | (13253) | (19403) (1384) (0.0468) | |
| Penalty I (4) | $2.2499\ldots 10^{-5}$ (5) | $2.2499\ldots 10^{-5}$ (4) | $2.2499\ldots 10^{-5}$ |
| | (7854) | (293609) (19379) (0.7656) | |
| Penalty II (4) | $9.3762\ldots 10^{-6}$ (1) | $9.3762\ldots 10^{-6}$ (2) | $9.3762\ldots 10^{-6}$ |
| | (5322) | (11056770) (710865) (65.8583) | |
| Osborne 1 (5) | $5.4648\ldots 10^{-5}$ (1) | $5.6507\ldots 10^{-5}$ (4) | $5.4648\ldots 10^{-5}$ |
| | (2790) | (2434886) (134400) (56.4898) | |

**Table 1** (*continued*)

| Test Function (*n*) | GNMa $\frac{(Acc.)(BestSolver)}{(FunctionEv.)}$ | MTNMa $\frac{(Accuracy)(BestSolver)}{(FunctionEv.)(SimplexEv.)(Time)}$ | Actual Minima |
|---|---|---|---|
| Brown −linear (5) | 0.0 (1) | $1.1044\ldots10^{-28}$ (5) | 0.0 |
| | (2788) | (18023) (920) (0.1093) | |
| Extended Rosenbrock (6) | $3.9443\ldots10^{-31}$ (1) | 0.0 (2) | 0.0 |
| | (7494) | (7742) (210) (0.0468) | |
| Watson (6) | $2.2876\ldots10^{-3}$ (1) | $2.2887\ldots10^{-3}$ (1) | $2.2876\ldots10^{-3}$ |
| | (5151) | (2831174) (123040) (150.0159) | |
| Brown almost linear (7) | $4.4373\ldots10^{-31}$ (3) | $3.1000\ldots10^{-26}$ (4) | 0.0 |
| | (11638) | (124461) (4520) (0.5203) | |
| Brown almost linear (7) | ∗ | 1.0000 (2) | 1.0000 |
| | ∗ | (152257) (5177) (0.5203) | |
| Quadratic (8) | 0.0 (1) | $3.0913\ldots10^{-320}$ (1) | 0.0 |
| | (39785) | (39149) (1410) (0.1240) | |
| Extended Rosenbrock (8) | $2.7523\ldots10^{-29}$ (1) | 0.0 (2) | 0.0 |
| | (19164) | (10144) (210) (0.0680) | |
| Variably dimensioned (8) | $8.0365\ldots10^{-30}$ (2) | 0.0 (5) | 0.0 |
| | (9336) | (5158) (164) (0.0468) | |
| Extended Powell singular (8) | $9.7234\ldots10^{-61}$ (1) | $4.9406\ldots10^{-324}$ (4) | 0.0 |
| | (20353) | (168349) (5190) (1.2031) | |
| Extended Rosenbrock (10) | $9.0484\ldots10^{-29}$ (1) | 0.0 (2) | 0.0 |
| | (36268) | (12546) (210) (0.0937) | |
| Penalty I (10) | $7.0876\ldots10^{-5}$ (2) | $7.6334\ldots10^{-5}$ (1) | $7.0876\ldots10^{-5}$ |
| | (25735) | (1987) (40) (0.0468) | |
| Penalty II (10) | $2.9411\ldots10^{-4}$ (2) | $2.9404\ldots10^{-4}$ (1) | $2.9366\ldots10^{-4}$ |
| | (51485) | (26235588) (526010) (142.4195) | |
| Trigonometric (10) | $4.4735\ldots10^{-7}$ (2) | 0.0 (5) | 0.0 |
| | (7253) | (13565) (320) (0.4218) | |
| Osborne 2 (11) | $4.0137\ldots10^{-2}$ (1) | $4.0137\ldots10^{-2}$ (5) | $4.0137\ldots10^{-2}$ |
| | (7381) | (16271) (391) (2.3126) | |
| Extended Powell singular (12) | $5.7700\ldots10^{-58}$ (1) | $6.4228\ldots10^{-323}$ (2) | 0.0 |
| | (50117) | (283723) (5770) (1.6563) | |
| Quadratic (16) | 0.0 (1) | 0.0 (2, 4) | 0.0 |
| | (112564) | (9420, 9398) (140) (0.0780) | |
| Quadratic (24) | $8.0493\ldots10^{-173}$ (1) | $5.0049\ldots10^{-280}$ (2, 4) | 0.0 |
| | (158849) | (105086) (1240) (0.7158) | |
| Variably dimensioned (36) | ∗ | $1.3353\ldots10^{-15}$ (2) | 0.0 |
| | ∗ | (34042) (160) (0.9626) | |
| Extended Rosenbrock (36) | ∗ | $4.9895\ldots10^{-29}$ (1) | 0.0 |
| | ∗ | (132771) (750) (3.1718) | |
| Discrete integral (50) | ∗ | $1.9158\ldots10^{-26}$ (1) | 0.0 |
| | ∗ | (40595) (200) (29.2100) | |

**Table 1** (*continued*)

| Test Function (*n*) | GNMa (*Acc.*)(*BestSolver*) (*FunctionEv.*) | | MTNMa (*Accuracy*)(*BestSolver*) (*FunctionEv.*)(*SimplexEv.*)(*Time*) | Actual Minima |
|---|---|---|---|---|
| Trigonometric (60) | * | | $2.8095\ldots10^{-18}$ (2) | 0.0 |
| | * | | (16471) (350) (51.1432) | |
| Extended Powell singular (60) | * | | $4.2421\ldots10^{-201}$ (4) | 0.0 |
| | * | | (1052158) (4290) (22.3512) | |
| Broyden tridiagonal (60) | * | | $3.7846\ldots10^{-27}$ (3) | 0.0 |
| | * | | (209538) (920) (5.7183) | |
| Broyden banded (60) | * | | $1.0733\ldots10^{-29}$ (1) | 0.0 |
| | * | | (95712) (390) (6.3707) | |
| Extended Powell singular (100) | * | | $6.7816\ldots10^{-315}$ (1) | 0.0 |
| | * | | (2753696) (6680) (109.2287) | |

**Notes.**
An asterisk (*) indicates that the GNMa has not been examined on particular test functions.

similar algorithms and help to allocate an accurate estimate of the computational budget for the compared algorithms. We choose to compare our proposed solution to GNMa (*Fajfar, Puhan & Burmen, 2017*) because GNMa is one of the best algorithms that utilizes the test functions of *More, Garbow & Hillstrom (1981)* and utilizes normalized data profile that involves one dimension (simplex evaluations). The GNMa generates solvers in a tree-based genetic programming structure. The population size is initialized to 200 and evaluated recursively to produce the evolving simplexes. The GNMa is implemented using twenty 2.66 Ghz Core i5 (four cores per CPU) machines (*Fajfar, Puhan & Burmen, 2017*). The authors assumed that a solution is acceptable if the fitness of the obtained solver is lower than $10^{-5}$. After running the computer simulation 20 times for 400 generations, five genetically evolved solvers successfully satisfied the condition of the fitness. The optimal solver is determined to be (genetic solver1).

Table 1 illustrates the produced results by MTNMa to cover the procedures for testing the reliability and robustness of the MTNMa. The results in this research are compared to the best-known relevant results from the literature presented by *Fajfar, Puhan & Burmen (2017)*. According to the definition of the normalized data profile (Eq. (21)), $f_L$ is required to be determined, which is the best obtained results by any of the individual solvers of the algorithms (GNMa and MTNMa). Therefore, Table 1 includes the best results of the GNMa obtained by any of the five genetic evolved solvers (the optimal genetic solver1 and the other four genetic solvers reported by *Fajfar, Puhan & Burmen (2017)*) to secure a fair comparison between GNMa and MTNMa. The GNMa is not an ensemble of the five evolved solvers and for this reason we utilize the high dimensional normalized data profiles to compare the MTNMa to the individual evolved solvers of the GNMa. Moreover, we compare the MTNMa to the one based on our previous publication in *Musafer & Mahmood (2018)*. The traditional triangular simplex of HNMa generates a simplex with specified edge length and direction that depends on the standard parameter values of $\delta_z$ and $\delta_u$, which is similar to solver 1. Table 1 also shows the dimensions of the test functions *n*, the number of simplex and function evaluations, and the actual minima known for the functions. In addition, the starting points for the test functions of *More, Garbow &*

*Hillstrom (1981)* are specified as part of the testing procedure so that the relevant algorithms can easily be examined and observed to validate whether the considered algorithms are tuned to a particular category of optimization problems or not. The other vertices can be either randomly generated (*Fajfar, Puhan & Burmen, 2017*) or produced using a specific formula such as the Pfefferś method (*Baudin, 2009*).

From the results given in Table 1, it can be seen that the proposed sequences of trigonometric simplex designs, in some cases, achieve a higher degree of accuracy for high dimensions than for less. For example, MTNMa performs better when optimizing Quadratic (16) as compared to Quadratic (8) in Table 1. In other cases, the MTNMa generates fewer simplexes to approximate a particular solution for high dimensions than for lower dimensions. For example, observe the number of simplex evaluations generated for Rosenbrock (6) compared to Rosenbrock (2) in Table 1. The behavior of the MTNMa in these problems is that when the dimensionality increases, the MTNMa manages to observe more patterns and find more combinations of the non-isometric features to form the reflected vertex. On the contrary, this is not the behavior of the GNMa, where the accuracy drops down and the algorithm performs a large number of simplex evaluations as it moves to higher dimensions. It can be observed also from Table 1 that the MTNMa was successful in following curved valleys functions such as Rosenbrock function. In addition, the test shows that the MTNMa is able to generate the same number of simplexes to reach the exact minimum for Rosenbrock (6, 8, and 10).

Thus testing MTNMa on much more complicated function such as Trigonometric (10) is useful because this function has approximately 120 sine and cosine functions added to each other. Even with the power of genetic programming, it is hard for the simplexes of the GNMa to progress in such an environment. However, since the proposed simplexes of the MTNMa have the angular rotation capability, they are capable of converging to minimums where amplitudes and angles are involved. Finally, we can see from Table 1 that the MTNMa can detect functions with multiple minimal values such as the Brown Almost Linear (7) function. In addition, the results indicate that the MTNMa outperforms the GNMa in terms of the accuracy tests for almost all high dimensional problems (more than or equal to 8).

Figure 3 contains four data profiles with different dimensions of performance metrics. One of the aims of utilizing various performance measures is to provide complementary information for the relevant solvers as the function of the computational budget. This is required to secure a fair comparison between the MTNMa and GNMa. As shown in Fig. 3–I, Fig. 3–II, and Fig. 3–IV the MTNMa needs to create 199 simplexes and 4,200 function evaluations to solve 100% of the problems at the level of accuracy $10^{-3}$. While the GNMa (genetic Solver1) needs to produce 2,700 simplexes to solve approximately 100% of the problems at this level of accuracy based on the reported results in *Fajfar, Puhan & Burmen (2017)*. Figure 3–III illustrates that the computational time takes about 1.3 s for the MHNMa to generate 199 simplex evaluations.

As it can be seen in Fig. 4–I, Fig. 4–II and Fig. 4–IV, solvers of the MTNMa require fewer number of simplex and function evaluations than solvers of the GNMa to solve roughly 100% of the problems. For example, with a budget of 200 simplex gradients, 10,225 function

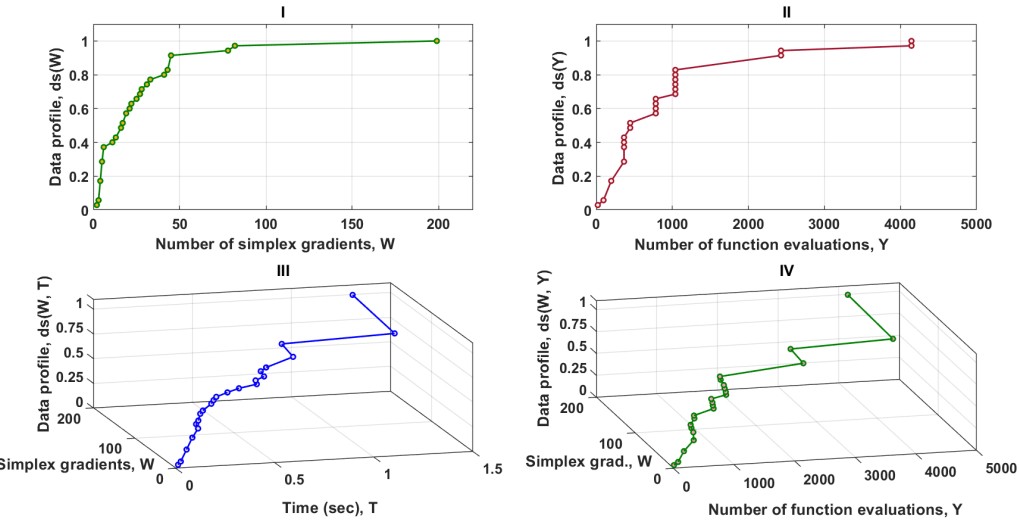

**Figure 3** **Data profiles for the MTNMa shown for ($\tau = 10^{-3}$).** I–Percentage of solved problems with respect to the number of simplex gradients (W), II–Percentage of solved problems with respect to the number of function evaluations (Y), III–Percentage of solved problems with respect to the number of simplex gradients (W) and the computer time (T), IV–Percentage of solved problems with respect to the number of simplex gradients (W) and the number of function evaluations (Y).

evaluations, and 2 s. Solvers of the MTNMa solve 100% of the problems at accuracy ($10^{-3}$), and solve almost 90% of the problems at accuracy ($10^{-5}$). This is a significant difference in performance. In addition, the computational complexity of the MTNMa solvers is not expensive as compared as the computational time and complexity required to evolve the GNMa solvers. The GNMa involves high computational overhead that comes from exchange vertices and features among the genetic simplexes and modernizing the current population with better offspring. The last major difference is that the optimization solutions of GNMa solvers in some functions are not able to satisfy Eq. (21) for this level of accuracy. For example, Trigonometric function (10) requires that the best possible reduction has to equal ($10^{-8}$), which is beyond the skills of any of the genetic solvers of GNMa.

From the sub-fig I, II and III given in Fig. 5, it can be seen that the MTNMa solves roughly 91% of the problems with a computational budget of 605 simplex gradients , 16271 function estimates, and 3.4 s. for the accuracy level of ($10^{-7}$). Another interesting observation on the data profiles shown in Figs. 4 and 5, is that the proposed algorithm tends to provide similar performance, as well as generate a moderate number of simplex and function evaluations to approximate solutions for the levels of accuracy ($10^{-5}$) and ($10^{-7}$). As a result, the use of data profiles that incorporate several performance metrics is essential to differentiate between similar algorithms, and provide an accurate estimate for allocating a computational budget that does not rely on a single dimension such as simplex gradients.

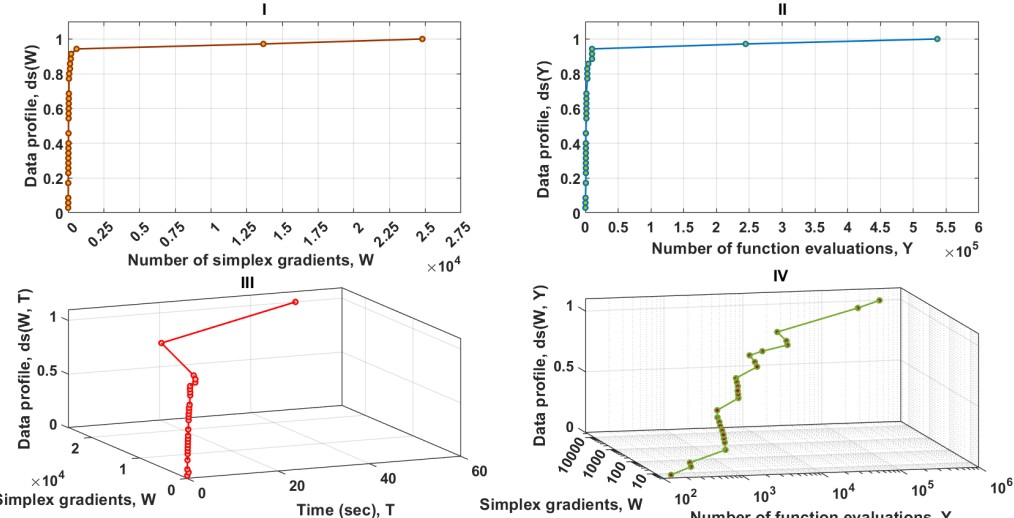

**Figure 4** **Data profiles for the MTNMa shown for ($\tau = 10^{-5}$).** I–Percentage of solved problems with respect to the number of simplex gradients (W), II–Percentage of solved problems with respect to the number of function evaluations (Y), III–Percentage of solved problems with respect to the number of simplex gradients (W) and the computer time (T), IV–Percentage of solved problems with respect to the number of simplex gradients (W) and the number of function evaluations (Y).

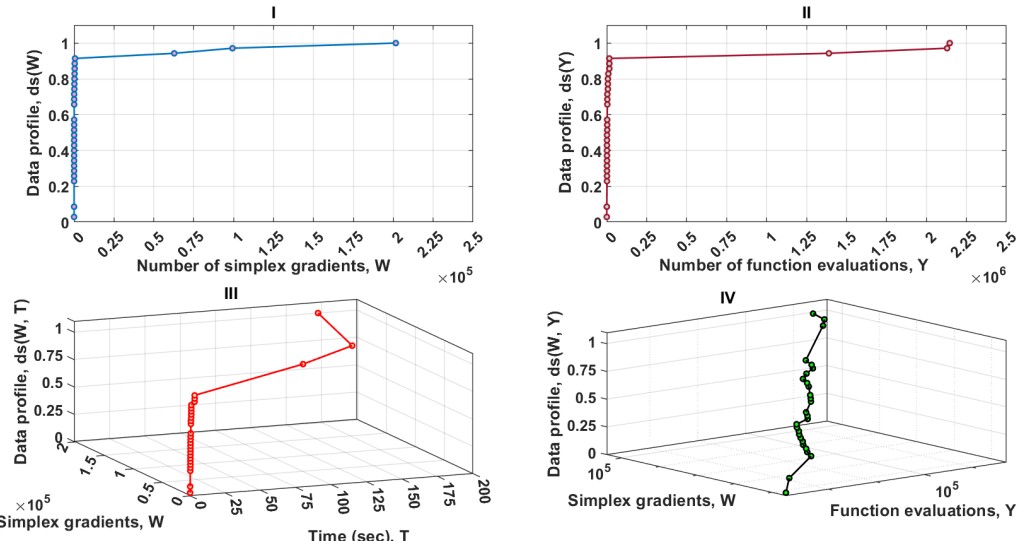

**Figure 5** **Data profiles for the MTNMa shown for ($\tau = 10^{-7}$).** I–Percentage of solved problems with respect to the number of simplex gradients (W), II–Percentage of solved problems with respect to the number of function evaluations (Y), III–Percentage of solved problems with respect to the number of simplex gradients (W) and the amount of computer time (T), IV–Percentage of solved problems with respect to the number of simplex gradients (W) and the number of function evaluations (Y).

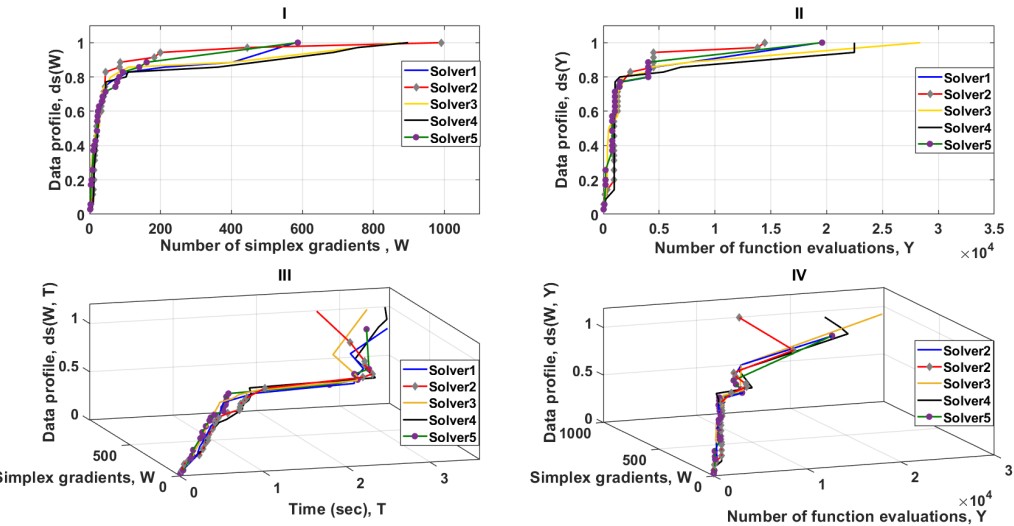

**Figure 6** **Data profiles for the five solvers shown for ($\tau = 10^{-3}$).** I–Percentage of solved problems with respect to the number of simplex gradients (W), II–Percentage of solved problems with respect to the number of function evaluations (Y), III–Percentage of solved problems with respect to the number of simplex gradients (W) and the amount of computer time (T), IV–Percentage of solved problems with respect to the number of simplex gradients (W) and the number of function evaluations (Y).

## Detailed analysis of the five solvers

We have conducted further tests by analyzing the five multi-directional trigonometric simplex solver designs. These reveal that higher dimensional data profiles are essential to deciding which solver should be used with a limited computational budget.

As shown in Fig. 6–part I, the dominant solver is 2 and tends to be faster than others for the first 400 simplex evaluations, solving almost 95% of the problems. In contrast, solvers (1 and 5) catch up after approximately 400 simplex evaluations, and outperform the others. The data profile of Fig. 6–I shows also that solvers (1 and 5) require significantly fewer number of simplex gradients than solver2 to solve 100% of the problems. Nevertheless, this significant difference in performance is not true when two performance metrics or more are used to examine the reliability of the solvers.

Figure 6–III illustrates, the cost unit per iteration (simplex evaluations (W) and time (T)) for solver 2 is less expensive than the other solvers. This forms a strong argument as to how a solver, in some cases, may require a larger number of simplex gradients but may have the potential to take less time to solve 100% of the test problems. Additional tests and analyzes shown in Fig. 6–II and Fig. 6–III, indicate the strength of combining metric measures in data profiles, forming a clear view that the cost unit (function evaluations (Y) and T) for solver 2 is much less expensive than the other solvers. Even if solver 2 requires more simplex gradient evaluations, it is still more reliable than the others. The results shown in Fig. 6–IV are fully consistent with the data profiles of Fig. 6–II and Fig. 6–III. Solver 2 stands out as being the best of the five solvers.

In this particular case, comparison of dimensions (W and Y) is useful for exploring how the number of active simplexes of solvers (1, 2 and 5) changes with respect to the number

of objective function evaluations. It is not obvious whether the overall performance of the solvers (1, 2, and 5) is almost entirely dependent on the number of objective function evaluations alone or not. If the number of function evaluations is the dominant dimension to achieve the presented results for Solver2, then the parameters (T and W) do not present independent dimensions and therefore T is dependent of Y in this particular case. This means that if the dimension T is removed from the profile, then remaining dimensions will show enough evidence to evaluate the five solvers.

To examine the parameters (T and W), we consider the observation of the relative performance of Solver3 and Solver4. The data profile as shown in Fig. 6–I indicates that Solver3 tends to produce less simplex evaluations than Solver4 to successfully solve the test problems. In contrast, the data profile in Fig. 6–II reveals that Solver4 needs to perform significantly less function evaluations than Solver3 to successfully solve the test problems. This can be seen in Fig. 6–III, where the data profile for the two dimensions (W for Y) is less computationally expensive for Solver4 than for Solver3. If we assume that the parameter T is dependent of Y, then the data profile shown in Fig. 6–IV should confirm that the cost unit (W and T) is less computationally expensive for Solver4 than for Solver3. Whereas, the data profile shows that the cost unit per iteration (W and T) for Solver3 is slightly less than the cost unit for Solver4. Therefore, T is independent of Y because there is an additional (non-constant) overhead associated with the relative complexity of the 5 MTNMa solvers that is independent of the number of function evaluations. The additional overhead comes from the exploration process around the neighborhood of the best result, which depends on how efficient a solver to move in a direction towards the optimum.

Solver3 requires higher function evaluations than Solver4, but takes less computational time to successfully solve the test problems. In this situation, Solver2 stands out as being the best of the five solvers because it requires fewer function evaluations and less computational time than the other solvers. This proves that the parameters (W, Y, and T) present independent dimensions for data profiling.

The number of CPUs (Z) was not examined in our evaluation of the MTNMa and is included in formula (22) for completeness. This dimension is significant if an optimiser is deployed in a distributed environment such as Amazon Web Services (AWS). In such a case, the number of nodes in the virtual cluster is an important aspect of the computational budget and the inclusion of Z assists in the allocation of optimal numbers of CPUs for different solvers and for specific levels of accuracy.

On a final note, the additional tests for examining data profiles on the five solvers of the MTNMa have confirmed that we need to define the normalized data profiles on the basis of a collection of performance measures. If the data profiles are defined for one dimension, then the accuracy of the profiles can be strongly biased when the numbers of function evaluations are independent of the other dimensions (simplex evaluations and computational time).

## CONCLUSION

In this work, we proposed five sequences of trigonometric simplex designs for high dimensional unconstrained optimization problems. In addition, each design extracts

different non-isometric reflections and performs a rotation determined by the collection of the non-isometric reflections. When executing multiple solvers simultaneously, a linear model with a window of size 10 samples is suggested as the criteria by which a solver is aborted or continued based on the direction vector of the window. We also showed in this research that using a data profile based only on the number of simplex gradients (one dimension) for allocation of the computational budget and examination of the relative performance of multiple solvers is not appropriate when simplex (W), function (Y), and time (T) evaluations present independent dimensions for data profile. Therefore, the definition of the suggested data profile has to involve different performance metrics. Then, the normalized data profile can be used not only to examine the efficiency and robustness of derivative free optimization algorithms but also to measure the relative computational time and complexity among the algorithms. Finally, the experimental results demonstrate that the MTNMa solvers outperform the GNMa solvers in terms of such data profiles that depend on different performance metrics for all levels of accuracy. In the future work, we will examine how reliable and robust MTNMa to the state-of-the-art DFO algorithms, such as the NOMAD software that is designed for difficult blackbox optimization problems (*Le Digabel, 2011*).

## APPENDIX

This section summarizes some of the common test functions designed for testing unconstrained optimization algorithms. The test functions are grouped according to their artificial landscapes into three classes: systems of nonlinear equations, nonlinear least squares, and unconstrained minimization. Let $f(x)$ be a nonlinear least squares problem whose terms exist in $f_1, f_2, \ldots, f_K$, then $f(x)$ is an unconstrained minimization problem such that

$$f(x) = \sum_{k=1}^{K} f_k^2(x) \tag{23}$$

If $K = n$, then the problem is a system of nonlinear equations and can be summarised in the next equation.

$$f_k(x) = 0, \quad 1 \leq k \leq n \tag{24}$$

And if $K > n$, then the optimal conditions for Eq. (23) are defined as a system of nonlinear equations such that

$$\sum_{k=1}^{K} \left( \frac{\partial f_k(x)}{\partial x_q} \right), \quad 1 \leq q \leq n \tag{25}$$

We follow a general format in the definition of the test functions to include the following elements such as name of function, description, standard starting point, and global minimum.

1. **Rosenbrock function** (*Rosenbrock, 1960*) $f_1(x) = 10(x_2 - x_1^2)$, $f_2(x) = (1 - x_1)$
   Description: The function is continuous, differentiable, non-separable, scalable,

non-convex, and unimodal and has a long valley with very steep walls and almost flat bottom (*More, Garbow & Hillstrom, 1981*).
Dimensions: $n = 2$, $K = 2$.
Standard starting point: $x_0 = (-1.2, \ 1)$.
Global minimum: $f(x) = 0$ at $(1, \ 1)$.

2. **Freudenstein and Roth function** (*Freudenstein & Roth, 1963*)
$f_1(x) = -13 + x_1 + ((5 - x_2)x_2 - 2)x2$, $f_2(x) = -29 + x_1 + ((1 + x_2)x_2 - 14)x2$
Description: The function is continuous, differentiable, non-separable, non-scalable, and multimodal, and contains a long shaped-valley, and is designed to have different sensitivities of the different variables.
Dimensions: $n = 2$, $K = 2$.
Standard starting point: $x_0 = (0.5, \ -2)$.
Global minimum: $f(x) = 0$ at $(4, 5)$, and $f(x) = 48.9842...$ at $(= 11.4125..., -0.8968...)$.

3. **Powell badly scaled function** (*Powell, 1970*)
$f_1(x) = 10^4 \cdot x_1 x_2 - 1$, $f_2(x) = e^{-x_1} + e^{-x_2} - 1.001$
Description: The function is continuous, differentiable, non-separable, non-scalable, and very flat near the global minimum point, and is used to test the optimization algorithm whether or not it can provide a sufficiently accurate estimate for the minimizer.
Dimensions: $n = 2$, $K = 2$.
Standard starting point: $x_0 = (0, \ 1)$.
Global minimum: $f(x) = 0$ at $(1.0981...10^{-5}, \ 9.1061)$.

4. **Brown badly scaled function** (*More, Garbow & Hillstrom, 1981*)
$f_1(x) = x_1 - 10^6$, $f_2(x) = x_2 - 2 \cdot 10^{-6}$, $f_3(x) = x_1 x_2 - 2$
Description: The function is continuous, non-convex, differentiable, and non-separable, and classified under valley-shaped optimization problems.
Dimensions: $n = 2$, $K = 3$.
Standard starting point: $x_0 = (1, \ 1)$.
Global minimum: $f(x) = 0$ at $(10^6, \ 2 \cdot 10^{-6})$.

5. **Beale function** (*Jamil & Yang, 2013*)
$f_1(x) = 1.5 - x_1(1 - x_2)$, $f_2(x) = 2.25 - x_1(1 - x_2)$, $f_3(x) = 2.625 - x_1(1 - x_2)$
Description: The function is continuous, differentiable, non-separable, non-scalable, and unimodal, and has sharp peaks at the corners.
Dimensions: $n = 2$, $K = 3$.
Standard starting point: $x_0 = (1, \ 1)$.
Global minimum: $f(x) = 0$ at $(3, 0.5)$.

6. **Jennrich and Sampson function** (*Jennrich & Sampson, 1968*)
$f_1(x), ..., f_k(x) = 2 + 2 - (e^{x_1} + e^{x_2}), ..., 2 + 2k - (e^{kx_1} + e^{kx_2})$
Description: The function is continuous, differentiable, non-separable, non-scalable, and multimodal.
Dimensions: $n = 2$, $K = 10$.
Standard starting point: $x_0 = (0.3, 0.4)$.
Global minimum: $f(x) = 124.362...$ at $(0.2578..., 0.2578...)$.

7. **Helical valley function** (*Fletcher & Powell, 1963*)

$$f_1(x) = 10(x_3 - 10 \cdot \theta(x_1, x_2)), \quad f_2(x) = 10\left(\sqrt{x_1^2 + x_2^2} - 1\right), \quad f_3(x) = x_3$$

$$\theta(x_1, x_2) = \begin{cases} \dfrac{1}{2\pi} tan^{-1}\left(\dfrac{x_2}{x_1}\right), & \text{if } x_1 > 0 \\ \dfrac{1}{2\pi} tan^{-1}\left(\dfrac{x_2}{x_1}\right) + 0.5, & \text{if } x_1 < 0 \end{cases}$$

Description: The function is continuous, differentiable, non-separable, scalable, and multimodal, and has a steep-sided helical valley in the direction of $x_3$ (*Figueroa & Schlick, 1992*).

Dimensions: $n = 2$, $K = 3$.

Standard starting point: $x_0 = (-1, 0, 0)$.

Global minimum: $f(x) = 0$ at $(1, 0, 0)$.

8. **Bard function** (*Bard, 1970*)

$$f_1(x), \ldots, f_k(x) = \vartheta_1 - \left(x_1 + \frac{1}{\varphi_1 \cdot x_2 + \varrho_1 \cdot x_3}\right), \ldots, \vartheta_k - \left(x_1 + \frac{k}{\varphi_k \cdot x_2 + \varrho_k \cdot x_3}\right) \text{ where } 1 \le k \le K,$$
$\varphi_k = 16 - k$, $\varrho_k = min(k, \varphi_k)$, and $\vartheta_k = 0.14$, 0.18, 0.22, 0.25, 0.29, 0.32, 0.35, 0.39, 0.37, 0.58, 0.73, 0.96, 1.34, 2.10, *and* 4.39.

Description: The function is continuous, differentiable, non-separable, non-scalable, and multimodal, and becomes flatter in the direction of $x_1$ when the other two parameters $x_2$ and $x_3$ decrease.

Dimensions: $n = 3$, $K = 15$.

Standard starting point: $x_0 = (1, 1, 1)$.

Global minimum: $f(x) = 8.2148\ldots10^{-3}$ at $(0.0824\ldots, 1.1332\ldots, 2.3434\ldots)$.

9. **Gaussian function** (*More, Garbow & Hillstrom, 1981*)

$$f_1(x), \ldots, f_k(x) = x_1 \cdot e^{\left(\frac{-x_2(\varphi_1 - x_3)^2}{2}\right)} - \vartheta_1, \ldots, x_1 \cdot e^{\left(\frac{-x_2(\varphi_k - x_3)^2}{2}\right)} - \vartheta_k \text{ where } 1 \le k \le K,$$
$\varphi_k = \frac{8-k}{2}$, and $\vartheta_k = 0.0009$, 0.0044, 0.0175, 0.0540, 0.1295, 0.2420, 0.3521, 0.3989, 0.3521, 0.2420, 0.1295, 0.0540, 0.0175, 0.0044, *and* 0.0009.

Description: The function is continuous, differentiable, non-separable, non-scalable, and multimodal. Dimensions: $n = 3$, $K = 15$.

Standard starting point: $x_0 = (0.4, 1, 0)$.

Global minimum: $f(x) = 1.1279\ldots10^{-8}$ at $(0.3989\ldots, 1.0000\ldots, 0)$.

10. **Meyer function** (*Meyer, 1970*)

$$f_1(x), \ldots, f_k(x) = x_1 \cdot e^{\left(\frac{x_2}{\varphi_1 + x_3}\right)} - \vartheta_1, \ldots, x_1 \cdot e^{\left(\frac{x_2}{\varphi_k + x_3}\right)} - \vartheta_k \text{ where } 1 \le k \le K,$$
$\varphi_k = 45 + 5k$, and $\vartheta_k = 34780$, 28610, 23650, 19630, 16370, 13720, 11540, 9744, 8261, 7030, 6005, 5147, 4427, 3820, 3307, *and* 2872.

Description: The function is continuous, differentiable, non-separable, and non-scalable, and represents a thermistor problem. The values of $\vartheta_k$ represent the resistance of a thermistor as a function of temperature $\varphi_k$.

Dimensions: $n = 3$, $K = 16$.

Standard starting point: $x_0 = (0.02, 4000, 250)$.

Global minimum: $f(x) = 87.9458\ldots$ at $(0.005609\ldots, 6181, 345.2)$.

11. **Gulf research and development function** (*Cox, 1969*)

$f_1(x),...,f_k(x) = e^{\left(-\frac{|\vartheta_1|x_3}{x_1}\right)} - \varphi_1,..., \ e^{\left(-\frac{|\vartheta_k|x_3}{x_1}\right)} - \varphi_k$ where $1 \le k \le K$, $\varphi_k = \frac{k}{100}$, and $\vartheta_k = 25 + (-50 \cdot ln(\varphi_k))^{\frac{2}{3}} - x_2$.

Description: The function is continuous, differentiable, non-separable, non-scalable, and multimodal, and has a very flat local minimum surrounded by a plateau, where the gradient is zero everywhere and the function equals 0.0385. This function is also known as the Weibull function.

Dimensions: $n = 3$, $K = 100$.

Standard starting point: $x_0 = (5, 2.5, 0.15)$.

Global minimum: $f(x) = 0$ at $(50, 25, 1.5)$.

12. **Box three-dimensional function** (*Box, 1966*)

$f_1(x),...,f_k(x) = e^{-\varrho_1 \cdot x_1} - e^{-\varrho_1 \cdot x_2} - x_3\left(e^{-\varrho_1} - e^{-10 \cdot \varrho_1}\right),..., \ e^{-\varrho_k \cdot x_1} - e^{-\varrho_k \cdot x_2} - x_3\left(e^{-\varrho_k} - e^{-10 \cdot \varrho_k}\right)$ where $1 \le k \le K$, and $\varrho_k = 0.1 \cdot k$.

Description: The function is continuous, differentiable, non-separable, and multimodal, and possesses an asymmetric curved valley.

Dimensions: $n = 3$, $K \ge n$.

Standard starting point: $x_0 = (0, 10, 20)$.

Global minimum: $f(x) = 0$ when $K = 3$, $(1, 10, 1)$, $(10, 1, -1)$, and $(x_1 = u, x_2 = u, 0)$ where $u \in R$.

13. **Powell singular function** (*Powell, 1962*)

$f_1(x) = x_1 + 10x_2, f_2(x) = \sqrt{5}(x_3 - x_4), f_3(x) = (x_2 - 2x_3)^2, f_4(x) = \sqrt{10}(x_1 - x_4)^2$

Description: The function is continuous, differentiable, non-separable, scalable, convex, and unimodal, and also known as a Powell quartic function (*Steihaug & Suleiman, 2013*). The function is difficult to minimize because the Hessian matrix at $f(x) = 0$ is doubly singular (*Brent, 2013*).

Dimensions: $n = 4$, $K = 4$.

Standard starting point: $x_0 = (3, -1, 0, 1)$.

Global minimum: $f(x) = 0$ at $(0, 0, 0, 0)$.

14. **Wood function** (*Colville, 2015*)

$f_1(x) = 10(x_2 - x_1^2), f_2(x) = 1 - x_1, f_3(x) = \sqrt{90}(x_4 - x_3^2), f_4(x) = 1 - x_3, f_5(x) = \sqrt{10}(x_2 + x_4 - 2), f_6(x) = \frac{x_2 - x_4}{\sqrt{10}}$

Description: The function is continuous, differentiable, non-separable, and multimodal, and is rather like Rosenbrock but with four variables and a quartic objective function. Many nonlinear programming codes fail to find the global minimum (*Brent, 2013*).

Dimensions: $n = 4$, $K = 6$.

Standard starting point: $x_0 = (-3, -1, -3, -1)$.

Global minimum: $f(x) = 0$ at $(1, 1, 1, 1)$.

15. **Kowalik and Osborne function** (*Kowalik & Osborne, 1968*)

$f_1(x),...,f_k(x) = \vartheta_1 - \frac{x_1(\varphi_1^2 + \varphi_1 \cdot x_2)}{(\varphi_1^2 + \varphi_1 \cdot x_3 + x_4)},..., \vartheta_k - \frac{x_1(\varphi_k^2 + \varphi_k \cdot x_2)}{(\varphi_k^2 + \varphi_k \cdot x_3 + x_4)}$ where $1 \le k \le K$, $\vartheta_k = $ (0.1957, 0.1947, 0.1735, 0.1600, 0.0844, 0.0627, 0.0456, 0.0342, 0.0323,) 0.0235, *and* 0.0246, and $\varphi_k = 4.0000, 2.0000, 1.0000, 0.5000, 0.2500, 0.1670, 0.1250, 0.1000,$ (0.0833, 0.0714, *and* 0.0625).

Description: The function is continuous, differentiable, non-separable, non-scalable, and multimodal, and arises from least squares fit of experimental data (*Winfield, 1973*).

Dimensions: $n = 4$, $K = 11$.

Standard starting point: $x_0 = (0.25, 0.39, 0.415, 0.39)$.

Global minimum: $f(x) = 3.0750...10^{-4}$ at $(0.1928..., 0.1912..., 0.1230..., 0.1360...)$.

16. **Brown and Dennis function** (*Brown & Dennis, 1971*)

    $f_1(x),...,f_k(x) = (x_1 + \varrho_1 \cdot x_2 - e^{\varrho_1})^2 + (x_3 + x_4 \cdot sin(\varrho_1) - cos(\varrho_1))^2$, ...,
    $(x_1 + \varrho_k \cdot x_2 - e^{\varrho_k})^2 + (x_3 + x_4 \cdot sin(\varrho_k) - cos(\varrho_k))^2$ where $1 \leq k \leq K$, and $\varrho_k = \frac{k}{5}$.

    Description: The function is continuous, differentiable, non-separable, non-scalable, and unimodal, and resembles a convex quadratic (*Yang, Ong & Jin, 2007*).

    Dimensions: $n = 4$, $K \geq n$.

    Standard starting point: $x_0 = (25, 5, -5, -1)$.

    Global minimum: $f(x) = 85822.2...$ when $(K = 20)$, and $x = -11.594..., 13.203..., -0.403..., (0.236...)$.

17. **Osborne 1 function** (*Osborne, 1972*)

    $f_1(x),...,f_k(x) = \vartheta_1 - (x_1 + x_2 \cdot e^{-\varrho_1 \cdot x_4} + x_3 \cdot e^{-\varrho_1 \cdot x_5})$, ...,
    $\vartheta_k - (x_1 + x_2 \cdot e^{-\varrho_k \cdot x_4} + x_3 \cdot e^{-\varrho_k \cdot x_5})$ where $1 \leq k \leq K$, $\varrho_k = 10 \cdot (k - 1)$, and $\vartheta_k = $
    0.844, 0.908, 0.932, 0.936, 0.925, 0.908, 0.881, 0.850, 0.818, 0.784, 0.751, 0.718, 0.685, 0.658, 0.628, 0.603, 0.580, 0.558, 0.538, 0.522, 0.506, 0.490, 0.478, 0.467, 0.457, 0.448, 0.438, 0.431, 0.424, 0.420, 0.414, 0.411, *and* 0.406.

    Description: The function is continuous, differentiable, non-separable, non-scalable, and multimodal, and has a very flat local minimum surrounded by a plateau, where the gradient is zero everywhere and the function equals 1.1060.

    Dimensions: $n = 5$, $K = 33$.

    Standard starting point: $x_0 = (0.5, 1.5, -1, 0.01, 0.02)$.

    Global minimum: $f(x) = 5.4648...10^{-5}$ at $x = (0.3754..., 1.9358..., -1.4647..., 0.01287...,0.02212...)$.

18. **Biggs EXP6 function** (*Biggs, 1971*)

    $f_1(x),...,f_k(x) = x_3 \cdot e^{-\varrho_1 \cdot x_1} - x_4 \cdot e^{-\varrho_1 \cdot x_2} + x_6 \cdot e^{-\varrho_1 \cdot x_5} - e^{-\varrho_1} + 5 \cdot e^{-10 \cdot \varrho_1} - 3 \cdot e^{-4 \cdot \varrho_1}$, ..., $x_3 \cdot e^{-\varrho_k \cdot x_1} - x_4 \cdot e^{-\varrho_k \cdot x_2} + x_6 \cdot e^{-\varrho_k \cdot x_5} - e^{-\varrho_k} + 5 \cdot e^{-10 \cdot \varrho_k} - 3 \cdot e^{-4 \cdot \varrho_k}$ where $1 \leq k \leq K$, and $\varrho_k = 0.1 \cdot k$.

    Description: The function is continuous, differentiable, non-separable, non-scalable, and multimodal, and involves $K$ exponential functions that all have steep valleys (*Figueroa & Schlick, 1992*).

    Dimensions: $n = 6$, $K \geq n$.

    Standard starting point: $x_0 = (1, 2, 1, 1, 1, 1)$.

    Global minimum: $f(x) = 0$ at $K = 13$, $x = (1, 10, 1, 5, 4, 3)$.

19. **Osborne 2 function** (*Osborne, 1972*)

    $f_1(x),...,f_k(x) = \vartheta_1 - \left(x_1 \cdot e^{-\varrho_1 \cdot x_5} + x_2 \cdot e^{-(\varrho_1 - x_9)^2 x_6} + x_3 \cdot e^{-(\varrho_1 - x_{10})^2 x_7} + x_4 \cdot e^{-(\varrho_1 - x_{11})^2 x_8}\right)$,
    ..., $\vartheta_k - \left(x_1 \cdot e^{-\varrho_k \cdot x_5} + x_2 \cdot e^{-(\varrho_k - x_9)^2 x_6} + x_3 \cdot e^{-(\varrho_k - x_{10})^2 x_7} + x_4 \cdot e^{-(\varrho_k - x_{11})^2 x_8}\right)$

    where $1 \leq k \leq K$, $\varrho_k = \frac{(k-1)}{10}$, and $\varrho_k = $ 1.366, 1.191, 1.112, 1.013, 0.991, 0.885, 0.831, 0.847, 0.786, 0.725, 0.746, 0.679, 0.608, 0.655, 0.616, 0.606, 0.602, 0.626, 0.651,

0.724, 0.649, 0.649, 0.694, 0.644, 0.624, 0.661, 0.612, 0.558, 0.533, 0.495, 0.500, 0.423, 0.395, 0.375, 0.372, 0.391, 0.396, 0.405, 0.428, 0.429, 0.523, 0.562, 0.607, 0.653, 0.672, 0.708, 0.633, 0.668, 0.645, 0.632, 0.591, 0.559, 0.597, 0.625, 0.739, 0.710, 0.729, 0.720, 0.636, 0.581, 0.428, 0.292, 0.162, 0.098, *and* 0.054.

Description: The function is continuous, differentiable, non-separable, non-scalable, and multimodal.

Dimensions: $n = 11$, $K = 65$. Standard starting point: $x_0 = (1.3, 0.65, 0.65, 0.7, 0.6, 3, 5, 7, 2, 4.5, 5.5)$.

Global minimum: $f(x) = 4.01377...10^{-2}$ at $x = 1.3097..., 0.4312..., 0.6335..., 0.5993..., 0.7532..., 0.9064..., 1.3654..., 4.8241..., 2.3989..., 4.5687..., 5.6753....$

20. **Watson function** (*Kowalik & Osborne, 1968*)

$f_1(x), ..., f_k(x) = \sum_{j=2}^{n}(j-1)x_j \vartheta_1^{j-2} - \left(\sum_{j=1}^{n} x_j \vartheta_1^{j-1}\right)^2 - 1, ..., \sum_{j=2}^{n}(j-1)x_j \vartheta_k^{j-2} - \left(\sum_{j=1}^{n} x_j \vartheta_k^{j-1}\right)^2 - 1$, *if* $1 \leq k \leq 29$ $f_k(x) = x_1$, *if* $k = 30$, and $f_k(x) = (x_2 - x_1^2 - 1)$, *if* $k = 31$ where $1 \leq k \leq K$, and $\vartheta_k = \frac{k}{29}$.

Description: The function is continuous, differentiable, non-separable, scalable, and unimodal. This minimization problem is ill-conditioned and difficult to solve (*Brent, 2013*).

Dimensions: $2 \leq n \leq 31$, $K = 31$.

Standard starting point: $x_0 = (0, ..., 0)$.

Global minimum: $f(x) = 2.2876...10^{-3}$ when $(n = 6)$ and $x = -0.015725..., 1.012435..., -0.232992..., 1.260430..., -1.513729..., 0.992996....$ $f(x) = 1.39976...10^{-6}$ when $(n = 9)$ and $x = -0.000015..., 0.999790..., 0.014764..., 0.146342..., 1.000821..., -2.617731..., 4.104403..., -3.143612..., 1.052627.$ $f(x) = 4.72238...10^{-10}$ at $(n = 12)$.

21. **Extended Rosenbrock function** (*Shang & Qiu, 2006*)

$f_k^1(x) = 10(x_{k+1} - x_k^2)$ *if* $(k \bmod 2) = 1$ $f_k^2(x) = (1 - x_{k-1})$ *if* $(k \bmod 2) = 0$ where $1 \leq k \leq K$.

Description: The function is continuous, differentiable, non-separable, scalable, multimodal, and non-convex (*Shang & Qiu, 2006*).

Dimensions: *n variable but even*, $K = n$.

Standard starting point: $x_0 = (-1.2, 1, ..., -1.2, 1)$.

Global minimum: $f(x) = 0$ at $x = (1, 1, ..., 1, 1)$.

22. **Extended Powell singular function** (*Steihaug & Suleiman, 2013*)

$f_k^1(x) = (x_k + 10x_{k+1}^2)$ *if* $(k \bmod 4) = 1$

$f_k^2(x) = \sqrt{5}(x_{k+1} - x_{k+2})$ *if* $(k \bmod 4) = 2$

$f_k^3(x) = (x_{k-1} - 2x_k)^2$ *if* $(k \bmod 4) = 3$

$f_k^4(x) = \sqrt{10}(x_{k-3} - x_k)^2$ *if* $(k \bmod 4) = 0$

where $1 \leq k \leq K$.

Description: The function is continuous, differentiable, non-separable, scalable, unimodal, and convex (*Steihaug & Suleiman, 2013*).

Dimensions: *n variable but a multiple of* 4, $K = n$.

Standard starting point: $x_0 = (3, -1, 0, 1, ..., 3, -1, 0, 1)$.

Global minimum: $f(x) = 0$ at $x = (0, 0, 0, 0, ..., 0, 0, 0, 0)$.

23. **Penalty I function** (*More, Garbow & Hillstrom, 1981*)

    $f_1(x), ..., f_k(x) = \sqrt{10^{-5}}(x_1 - 1), ..., \sqrt{10^{-5}}(x_k - 1), \quad if \ (1 \leq k \leq K - 1)$

    $f_k(x) = \left( \sum_{j=1}^{n} x_j^2 \right) - \frac{1}{4}, \quad if \ (k = K)$

    where $1 \leq k \leq K$.

    Description: The function is continuous, differentiable, non-separable, ill-conditioned, and difficult to solve.

    Dimensions: *n variable*, $K = n + 1$.

    Standard starting point: $x_0 = (1, 2, 3, 4)$ when $n = 4$, $x_0 = (1, 2, 3, 4, 5, 6, 7, 8, 9, 10)$ when $n = 10$.

    Global minimum: $f(x) = 2.2499...10^{-5}$ when $(n = 4)$.

    $f(x) = 7.0876...10^{-5}$ when $(n = 10)$.

24. **Penalty II function** (*More, Garbow & Hillstrom, 1981*)

    $f_k(x) = (x_k - 0.2), \quad if \ (k = 1)$

    $f_k(x) = \sqrt{10^{-5}} \left( e^{\frac{x_k}{10}} + e^{\frac{x_{k-1}}{10}} - e^{\frac{k}{10}} - e^{\frac{k-1}{10}} \right), \quad if \ (2 \leq k \leq \frac{K}{2})$

    $f_k(x) = \sqrt{10^{-5}} \left( e^{\frac{x_{k-n+1}}{10}} - e^{\frac{1}{10}} \right), \quad if \ (\frac{K}{2} < k \leq K - 1)$

    $f_k(x) = \left( \sum_{j=1}^{n} (n - j + 1) x_j^2 \right) - 1, \quad if \ (k = K)$

    where $1 \leq k \leq K$.

    Description: The function is continuous, differentiable, non-separable, ill-conditioned, and difficult to solve.

    Dimensions: *n variable*, $K = 2n$.

    Standard starting point: $x_0 = (0.5, ..., 0.5)$

    Global minimum: $f(x) = 9.3762...10^{-6}$ when $(n = 4)$.

    $f(x) = 2.9366...10^{-4}$ when $(n = 10)$.

25. **Variably dimensioned function** (*More, Garbow & Hillstrom, 1981*)

    $f_k(x) = (x_k - 1), \quad if \ (1 \leq k \leq K - 2)$

    $f_k(x) = \sum_{j=1}^{n} j(x_j - 1), \quad if \ (k = K - 1)$

    $f_k(x) = \left( \sum_{j=1}^{n} j(x_j - 1) \right)^2, \quad if \ (k = K)$

    where $1 \leq k \leq K$.

    Description: The function is continuous, differentiable, non-separable, and multimodel. The solution space is crossed flat area like U-curve (*Tippayawannakorn & Pichitlamken, 2013*).

    Dimensions: *n variable*, $K = n + 2$.

    Standard starting point: $x_0 = (1 - \frac{i}{n}, ...)$, where $(1 \leq i \leq n)$.

    Global minimum: $f(x) = 0$ at $x = (1, ..., 1)$.

26. **Trigonometric function** (*More, Garbow & Hillstrom, 1981*)

    $f_k(x) = n - \sum_{j=1}^{n} cos(x_j) + k(1 - cos(x_k)) - sin(x_k)$

    where $1 \leq k \leq K$.

    Description: The function is continuous, differentiable, non-separable, scalable, and multimodel, and difficult to converge to the global minimum (*Tippayawannakorn & Pichitlamken, 2013*).

    Dimensions: *n variable*, $K = n$. Standard starting point: $x_0 = (\frac{1}{n}, ..., \frac{1}{n})$.

Global minimum: $f(x) = 0$.

27. **Brown almost linear function** (*Brown, 1969*)

    $f_k(x) = x_k + \sum_{j=1}^{n} x_j - (n+1), \quad if \ (1 \le k \le K-1)$

    $f_k(x) = \left(\prod_{j=1}^{n} x_j\right) - 1, \quad if \ (k = K)$

    where $1 \le k \le K$.

    Description: The function is continuous, differentiable, non-separable, scalable, and unimodel.

    Dimensions: *n variable*, $K = n$. Standard starting point: $x_0 = (0.5, \ldots, 0.5)$.

    Global minimum: $f(x) = 0$ at $x_0 = (\varrho, \ldots, \varrho, \varrho^{1-n})$, where $\varrho$ *satisfies* $(n\varrho^n - (n+1)\varrho^{n-1})$. $f(x) = 1$ at $x_0 = (0, \ldots, 0, n+1)$.

28. **Discrete boundary value function** (*More & Cosnard, 1976*)

    $f_k(x) = 2x_1 - x_0 - x_2 + \frac{\varrho_1^2}{2}(x_1 + 1 \cdot \varrho_1 + 1)^3, \ldots, \ 2x_k - x_{k-1} - x_{k+1} + \frac{\varrho_k^2}{2}(x_k + k \cdot \varrho_k + 1)^3$

    where $1 \le k \le K$, $\varrho_k = \left(\frac{1}{n+1}\right)$, and $x_0 = x_{K+1} = 0$.

    Description: The function is continuous, differentiable, non-separable, non-scalable, and unimodel.

    Dimensions: *n variable*, $K = n$.

    Standard starting point: $x_0 = (1 \cdot \varrho_1(1 \cdot \varrho_1 - 1), \ldots, \ k \cdot \varrho_k(k \cdot \varrho_k - 1))$.

    Global minimum: $f(x) = 0$.

29. **Discrete integral function** (*More & Cosnard, 1976*)

    $f_k(x) \ =$
    $x_1 + \frac{\varrho_1}{2}\left((1 - 1 \cdot \varrho_1)\sum_{j=1}^{n} 1 \cdot \varrho_1(x_j + j \cdot \varrho_1 + 1)^3 + 1 \cdot \varrho_1 \sum_{j=1+1}^{n}(1 - j \cdot \varrho_1)(x_j + j \cdot \varrho_1 + 1)^3\right)$,
    $\ldots, \ x_k + \frac{\varrho_k}{2}\left((1 - k \cdot \varrho_k)\sum_{j=k}^{n} k \cdot \varrho_k(x_j + j \cdot \varrho_j + 1)^3 + k \cdot \varrho_k \sum_{j=k+1}^{n}(1 - j \cdot \varrho_j)(x_j + j \cdot \varrho_j + 1)^3\right)$

    where $1 \le k \le K$, $\varrho_k = \left(\frac{1}{n+1}\right)$, and $x_0 = x_{K+1} = 0$.

    Description: The function is continuous, differentiable, non-separable, non-scalable, and unimodel.

    Dimensions: *n variable*, $K = n$.

    Standard starting point: $x_0 = (1 \cdot \varrho_1(1 \cdot \varrho_1 - 1), \ldots, \ k \cdot \varrho_k(k \cdot \varrho_k - 1))$.

    Global minimum: $f(x) = 0$.

30. **Broyden tridiagonal function** (*Broyden, 1965*)

    $f_k(x) = x_0 - (3 - 0.5x_1)x_1 + 2x_2 - 1, \ldots, \ x_{k-1} - (3 - 0.5x_k)x_k + 2x_{k+1} - 1$

    where $1 \le k \le K$, and $x_0 = x_{K+1} = 0$.

    Description: The function is continuous, differentiable, separable, scalable, and multimodel.

    Dimensions: *n variable*, $K = n$.

    Standard starting point: $x_0 = (-1, \ldots, -1)$.

    Global minimum: $f(x) = 0$.

31. **Broyden banded function** (*Broyden, 1971*)

    $f_k(x) = (1 + x_1^2)x_1 + 1 - \sum_{j=-4; \ j \ne 1}^{2}(1 + x_j)x_j, \ldots, \ (1 + x_k^2)x_k + 1 - \sum_{j=k-5; \ j \ne k}^{k+1}(1 + x_j)x_j$

    where $1 \le k \le K$, and $(x_k = 0)$, *if* $k \le 0$, *or* $k > K$.

    Description: The function is continuous, differentiable, separable, scalable, and multimodel.

    Dimensions: *n variable*, $K = n$.

Standard starting point: $x_0 = (-1, \ldots, -1)$.
Global minimum: $f(x) = 0$.

### Funding
The authors received no funding for this work.

### Competing Interests
Hassan Musafer is employed by CDPHP.

### Author Contributions
- Hassan Musafer conceived and designed the experiments, performed the experiments, analyzed the data, performed the computation work, prepared figures and/or tables, authored or reviewed drafts of the article, and approved the final draft.
- Emre Tokgoz conceived and designed the experiments, analyzed the data, authored or reviewed drafts of the article, and approved the final draft.
- Ausif Mahmood conceived and designed the experiments, analyzed the data, authored or reviewed drafts of the article, and approved the final draft.

### Data Availability
The scripts are written in C# and are available in the Supplemental Information.

### Supplemental Information
Supplemental information for this article can be found online at http://dx.doi.org/10.7717/peerj-cs.960#supplemental-information.

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
