# Peer review of "High-dimensional normalized data profiles for testing derivative-free optimization algorithms"

_PeerJ Computer Science, doi:10.7717/peerj-cs.960_

## Round 0.1 · original submission · Major Revisions

Both reviewers provided very detailed comments and recommended major revisions.

Reviewer 1 ·

Basic reporting

- Mathematical notation in page 2 is not standard in the DFO literature. Please use widely-used mathematical notation.

- Capitalize the first letter in "equation"

- "The HNM algorithm proved to deliver better performance than the
traditional NM algorithm, represented by a famous Matlab function, known as ”fminsearch” Musafer and Mahmood (2018)". fminsearch has not been proposed by Musafer and Mahmood (2018), it is the classic NM algorithm.

- "From the table, ...". Please provide references to all figures/tables.

- The English language can be improved. Some examples where the language could be improved include lines 222 and 292 – the current phrasing makes
comprehension difficult.

- A readme file is needed in the Supplemental_File.zip in order to be able to find out where to find the codes and problem files.

Experimental design

- "The standard parameter values for usual delta d_u and for zero term delta d_z are chosen 0.05 and 0.00025 respectively". Please justify.

- The test functions have not been presented in the text. More importantly, all these problems have only a few variables. The authors need to consider much larger problems.

- What's the purpose of the computational study? Why not compare against efficient DFO algorithms (e.g., NOMAD)?

Validity of the findings

No comment

Additional comments

The paper is interesting. However, I am confused about its scope. Looking at the title of the paper (High-dimensional normalized data profiles for testing derivative-free optimization algorithms), I would expect that the authors propose a new tool (similar to data profiles) for comparing DFO algorithms. However, this is not the case since the authors are proposing five sequences of trigonometric simplex designs for high dimensional unconstrained optimization problems. Therefore, the authors need to focus on this and only present normalized data profiles as a mean to compare their algorithm. Thus, I believe that the authors need to restructure their paper in order to make this clear. Finally, the computational results are weak. The authors need to consider larger problems and other DFO algorithms to compare with.

Other issues

- Mathematical notation in page 2 is not standard in the DFO literature. Please use widely-used mathematical notation.

- Capitalize the first letter in "equation"

- "The standard parameter values for usual delta d_u and for zero term delta d_z are chosen 0.05 and 0.00025 respectively". Please justify.

- "The HNM algorithm proved to deliver better performance than the
traditional NM algorithm, represented by a famous Matlab function, known as ”fminsearch” Musafer and Mahmood (2018)". fminsearch has not been proposed by Musafer and Mahmood (2018), it is the classic NM algorithm.

- The test functions have not been presented in the text. More importantly, all these problems have only a few variables. The authors need to consider much larger problems.

- What's the purpose of the computational study? Why not compare against efficient DFO algorithms (e.g., NOMAD)?

- "From the table, ...". Please provide references to all figures/tables.

- The English language can be improved. Some examples where the language could be improved include lines 222 and 292 – the current phrasing makes
comprehension difficult.

- A readme file is needed in the Supplemental_File.zip in order to be able to find out where to find the codes and problem files.

·

Basic reporting

The prose of the article requires significant revision in order for it to meet the requirement for use of clear and unambiguous English. There are frequent grammatical errors and use of terms which are, to my experience, non-standard with respect to the common language of the mathematical sciences. A non-exhaustive list of some of the errors with proposed corrections is included in the “General comments for the author”.

The structure of the article also requires revision. For example, the first two paragraphs of the Discussion (starting Line 251) introduce the Genetic Nelder Mead (GNM) algorithm and provide the only summary of the author’s Multidirectional Trigonometric Nelder Mead (MTNM) algorithm. The Discussion section should be limited to the interpretation of the author’s work, in the context of previously presented background information, as much as possible.

As the GNM algorithm is a core component of the author’s evaluation of the MTNM algorithm and the proposed data profile (Equation (25)), I believe that it should be more prominently introduced in the article’s introduction. References should also be provided to justify the choice of the GNM algorithm over other Nelder Mead variants for use as a point of comparison. References should also be provided for the sentence starting on Line 220.

The minimisation example starting on Line 49 requires further explnation. While I do find the chosen topic (surgical grafts in cardiology) to be interesting, it is not clear with respect to what variables the objective function is defined.

There are a number of small issues with the Figures and Table presented in this article:
• Figure 1: Sublabel “A” has an identical to the variable “A” present in Figure 1 A and B. The parameters $d$, $d_l$ and $Th$ are not defined. The axes of the plot Figure 1 A are not labelled.

• Table 1 performs a comparison which I do not believe to be an accurate communication of the work of Fajfar et al (see Section 2).

• Figure 2, 3, 4 and 5: As data profiler variables are defined in the accompanying description, the axes of the plots may use ds(*), W, Y and T (sec) as the axes labels (suggestion). It is not clear how parameter Y is an “estimate” of the objective function, as the function should be computed up to the limits of numerical precision at each point. I suspect “estimates” refers to the number of “evaluations”.

• Figure 5 A and B: It is difficult follow the lines for each solver without zooming in considerably.

The authors have provided code for 6 of the 36 functions comprising test set proposed by More´ et al. I was able to execute this code to verify this limited subset of the author’s results. However, code should be provided for the complete suite of test functions.

Finally there are a number of, mostly small, mathematical omissions or errors:

• Line 51: The domain and range of $f$ is not defined. In the context of the Nelder Mead algorithm, the generalised objective function is typically $f: \mathbb{R}^n \rightarrow \mathbb{R}$ such that for, $f(x)$, $x \in \mathbb{R}^n $.

• Line 101 presents an ordering of vertex pairs, this is incorrect in light of the ordering described on the previous line. The ordering should be notated in a manner equivalent to: $A=f(v_1) < B=f(v_2) < C = f(v_3)…$

• Line 152: If the values for $delta_u$ and $\delta_z$ have been arrived at through the authors’ practical experience, they should state so, if there is a formal or historical justification for these parameter values, it should be explained.

• Equation (25): Parameter “Z” refers to the “number of machines”. The term “number of machines” is ambiguous. I assume, however, that the authors are referring to the number of CPU cores.

Experimental design

The author’s present two original contributions:

1. Firstly, a modified version of the Nelder Mead algorithm, the MTNM algorithm, utilizes a set of 5 distinct solvers which work independently to identify the objective function minimum. These simplexes introduce different rotational ‘shifts’ through modification of the functions governing the placement of new simplex vertices at each iteration. An early stopping criterion is used to abort solvers which are failing to converge. The MTNM algorithm is shown to outperform solvers developed via the GNM algorithm in some cases. Of particular note is its performance in high dimensions, for example, when minimising the Quadratic (24) problem.

2. Secondly, a higher dimensional data profiler function by which to evaluate the performance of two dissimilar solvers operating on the same problem set. It introduces dimensions associated with wall time, number of simplex evaluations and number of CPUs.

However, there are a number of significant issues with the experimental design:

• It is not clear why the GNM algorithm has been chosen as a point of comparison. In the referenced paper, Fajfar et al outline a generic algorithm (GNM) for the evolution (or “hyper-optimisation”) of alternative Nelder Mead simplexes. They arrived at 5 candidates and present a single “optimal solver” (solver 1). Conversely, the authors propose an algorithm that uses an ensemble of 5 solvers.

• Table 1 implies that the GNM algorithm is an ‘ensemble’ Nelder Mead solver when that is not the case. This also occurs on Line 306 where it is stated that the GNM algorithm requires about 12 hours to complete using 20 CPU cores. These values are taken from the time reported by Fajfar et al for completion of their genetic evolution algorithm, not optimisation of the standard test problems of More´ et al.

• The data presented by Fajfar et al does not allow for a comparison of their solvers with the MTNM algorithm using the author’s defined data profile.

• The addition of the T, W and Z parameters require further justification, as it stands, I have the following concerns:

o The Nelder Mead, and modified Nelder Mead, algorithms presented in these articles require $n + 1$ function evaluations per iteration, where $n$ is the number of objective function parameters. The early stopping test employed by the MTNM algorithms results in a meaningful difference between the number of simplex evaluations and function evaluations, but this is certainly not the case in general.

o While comparison of $W$ and $Y$ is perhaps useful for exploring how the number of ‘active’ MTNM simplexes changes with respect to the number of objective function evaluations, the overall performance of the algorithm is still almost entirely dependent on the number of objective function evaluations alone. Dependence on $W$ could only arise through the use of simplexes with distinct algorithmic complexity. If there is a measurable difference in the algorithmic complexity of the 5 MTNM solvers, the authors should demonstrate this by way of proofs or numerical experiments.

o The algorithm wall-time, $T$, is not generalizable due to its dependence on the particular computer hardware used and the state of the host computer. Furthermore, for $T$ to be independent of $Y$, there must exist a significant (non-constant) overhead associated with the solvers that are independent of the number of function evaluations. However, as any such time overhead will necessarily result from the algorithmic complexity of the simplex construction step (see above), the $T$ and $W$ parameters do not seem to present independent dimensions for data profiling.

o The number of CPUs, $Z$, was not used by the authors in their evaluation of the MTNM algorithm and I am unsure of how it relates to the NM algorithm in general. The most common target for parallelization techniques in the context of numerical optimisation problems is the objective function as it is (almost always) the most computationally expensive step. For $Z$ to be relevant the authors should present an optimisation algorithm that follows an alternative parallelization scheme.

Validity of the findings

The authors present interesting results with respect to the performance of the MTNM solver, however, exploration of these results is hampered by issues in the experimental design (see Section 2). Unforteuently, it is not possible to easily validate all of the author’s findings and reliance on externally sourced data (from the referenced paper by Fajfar et al) is an additional barrier to replication.

Additional comments

Below are some suggested grammatical corrections and comments relating to the prose of the submitted article:

Line 18: “…through an angle <that> designates…”

Line 37: “<We are> motived by…”

Line 47: “<It seems a fact> that if a problem can be described as a mathematical expression, then at some point we will be <required> to minimize it or tune <its> independent variables at a <minimum>”.

Line 59: “…reflections over the <changing> landscape of <the> mathematical problems until the coordinates of the minimum point <can> be obtained…”

Line 66: “…the simplex becomes <increasingly> distorted <with each iteration>, generating different geometrical formations that are less effective <than> the <original> simplex design.”

Line 69: “…the sequence of simplexes <to> converge to a non-stationary point.”

Line 75: “The rest of this paper is organised as follow<s>.” “presents the theory of <the> sequential design of trigonometric Nelder-Mead algorithms<s>…”

Line 76, 97 Figure 1 A: “…the vector theory” -> “vector theory”

Line 78: “…and presents <the> multidimensional…”

Line 83: “…the theory of <the> Hassan…”

Line 86: “<This is different from the> traditional NM algorithm…” (suggestion)

Line 93: “…the generat<ed> sequence of triangular simplexes are guaranteed not only have different shapes but <to> also they have different directions<.> <These characteristics lead to> better performance than the traditional hyperplanes simplex.” “ Note that <to> find the reflected point D we add the vectors $H$ and $d$<, as shown in Figure 1 A.> ”

Line 106: “Now is we consider two combinations or more”: It is not clear what sort of combinations we are considering.

Line 107: “..multiple components of <the> triangular HNM simplex…”

Line 114: “In fact, the HNM algorithm is designed to deform its simplex in a way
that introduces nonlinear movement, by incorporating a rotation<al> <shift at> each iteration.” (suggestion)

Line 115: “After all axial components of the Th point are updated to see whether the new Th is better than C to be replaced or not.”

The above sentence and, possibly the one before, would be better positioned before the concluding paragraph as they outline key aspects to the HNM algorithm.

Line 122: “…approximation to a solution, generating different geometrical configurations…”

Line 123: “…build the initial simplex with <edges of > equal length …”

Line 129: “The risk is that<,> if the initial simplex is perpendicular to an optimal solution, then the algorithm…”

Line 132: “…initial simplex with the NM method…” (suggested)

Line 134: “…the generat<ed> sequence of…”

Line 134: “One of the problems is that the generating sequence of simplexes should be consistently scaled with respect to the best vertex obtained and restart the simplex around the best observed results.”

This sentence is difficult to parse as “One of the problems” is closely followed by “should be”. The first phase is a negative imperative, while the second phrase is a positive imperative. A clearer sentence could start “One of the problems is that the generated sequence of simplexes is not scaled with respect to the best vertex…” or, perhaps, “Ideally the generated sequence of simplexes should be scaled with respect to the best vertex…”.

Line 138: “”Alternatively, the most popular way to initialize a simplex is Pfeffers´ method <(Baudin, 2009).> <This approach scales> the initial simplex <based on the starting point, $X_0$>. The initial vertex is set to v1 = x0, and the remaining vertices <are> obtained as follows,”

The paragraph starting Line 141: Please present a brief argument as to why Equation (15) leads to improved starting points. The last sentence does not deductively follow from the Equation definition.

Line 145: “On contrary, our solution depends on allowing the components of the reflected vertex to <transform non-uniformly with each iteration of the simplex>.

Line 154: New paragraph at “In this test, we are more interested in launching multiple sequences of trigonometric simplex designs with different…”

Line 158: “This procedure reinforces the standard Pfeffers´ method of creating a simplex with additional designs.”

The above sentence could be safely removed.

Line 158: I suggested that the sentence starting on this line, and the one following, be reworked.

The first sentence makes a definite claim as to the effectiveness of the multidirectional simplexes without providing justification, it is also not clear what “effective” means in the context. Finally, instead of “mathematical landscape”, I suggest the term “solution space” or “the domain of the objective function”.

In the following sentence, it is not clear (to me) if the proposed methods seek to rotate the simplexes such that they have a unique starting direction or a unique direction with each iteration.

For example, see the following rephrasing:

“<We will demonstrate that our> proposed multidirectional simplexes <converge to a minimum with a smaller> number of simplex gradient estimates <as compared to the previously discussed methods of simplex design>. <Key to this outcome is the> shifting <of> points associated with the highest, the second-highest, or the lowest values of the objective function with a displacement that <generates simplexes with> distinct directions.”

Line 163: One of the potential designs <multiplies> the odd<-indexed> variables of odd-<indexed> vertices by (-1)

Line 167: “…even components of x0,”

Line 168: “…specified edge lengths and orientation…” All simplex edge lengths and orientations are specified in some manner. Are the specifications those goals outlined in the paragraph starting on Line 145?

Line 169: “…by multiplying (-1) the parameters…” Is the (-1) a typo?

“usual delta” and “zero term delta” -> “$\delta_u$ and $\delta_0”

Line 170: “…the absolute value <of the components of $x_0$ and> and subtracting or adding to adjust the simplex orientation…”

Line 171: “…and can be modified as needed…” By what criteria are the position vectors added or subtracted?

Line 173: “…as well as the slope…” The slope with respect to what function and what point points?

“Therefore, a window of size 10 points…” The choice of 10 points does not deductively follow from the previous sentence, why 2, 4, or 100? If the window size is a hypermeter derived from your practical experience please make that choice clear.

Line 176: “…the direction vector, which designates the direction of the simplex…” -> “the direction of the simplex”

Line 178: “For the particular <case>,”


Line 189: Non-deductive therefore. The phrase "Therefore" should only when drawing a logical link between ideas.

Line 190: “tools” -> “metrics” or “figures of merit”, “similar algorithms,” -> “the considered algorithms.”

“Which are summarized as follows: <the> accuracy of the algorithm compared to the actual minima, <the wall-time to convergence>, the number of function evaluations, the number of simplex evaluations, and identification of the best sequence of trigonometric simplex design.

Line 194 to 200: “The main reason for designing the set of functions”: It is not clear how the set of functions relates to the previously mentioned testing guidelines. I suggest that the authors make the link explicit (e.g. “These guidelines utilize a set of functions…) or place this discussion after the subsequent sentence.

Line 207: What is “short-term” behaviour in this context?

Line: “launch” -> “launches”

Line 209: It is not clear to me how the individual solvers “cooperate” to find an optimal solution. As it stands I do not think they share information beyond their common starting point, $x_0$, is this correct?

Line 233: “All computational experiments were carried out on an i5 CPU (4 Cores at1.8 GHz) and 4 GB of RAM. We used C# to implement the MTNM algorithm and carry out the experiments.”

---

## Round 0.2 · Major Revisions

Please address the issues raised especially by Reviewer 1. Please also carefully proofread the manuscript and write more clearly as suggested by Reviewer 2.

Reviewer 1 ·

Basic reporting

The authors have addressed all my comments.

Experimental design

The authors addressed most of my comments. However, they did not address the most important two issues:

- The test functions have not been presented in the text. More importantly, all these problems have only a few variables. The authors need to consider much larger problems.
The authors state in their response that they want to be consistent with the literature in terms of the number of variables and use the same total number of variables of in order to be able to compare computational results. It is OK to include these problems in order to compare the results with the literature. However, it would add more value to the paper to also include larger problems.

- Why not compare against efficient DFO algorithms (e.g., NOMAD)?
The authors state that they choose GNMa because GNMa is one of the best algorithms that utilizes the test functions of (Moré et al., 1981) and utilizes normalized data profiles that involves one dimension (simplex evaluations). It is OK to use GNMa for the aforementioned reasons. However, why not use the state-of-the-art DFO algorithms, like NOMAD?

Validity of the findings

The authors have addressed all my comments.

Additional comments

The authors have addressed all my comments.

·

Basic reporting

The overall prose of the article has improved but significant proofreading and revision of some paragraphs are required in order to meet the criteria of clear and unambiguous professional English. Identified typos and suggested re-phrasing are included in 'Additional comments', however, I am certain that the list is non-exhaustive.

Experimental design

No comment.

Validity of the findings

No comment.

Additional comments

The experimental design is outlined clearly and the authors' original contributions are clearly defined. The authors introduce the Multidirectional Trigonometric Nelder Mead (MTNM) algorithm, which utilises an ensemble of five simplexes developed through variation to the simplex of the Hassan Nelder Mead Algorithm.

The performance of this algorithm is assessed against the Genetic Nelder Meader (GNM) algorithm, which provides a single simplex derived through genetic programming. To compare their ensemble solver to this algorithm the authors proposed a high-dimensional normalised data profile that takes into account the number of cost-function variables, cost function evaluations, simplex evaluations, wall-time and CPU count.

The authors provide sufficient background and references, have addressed previously identified issues in their diagrams and have shared the code needed to reproduce their original numerical results. The article is self-contained, presents results relevant to the hypotheses and includes clear definitions of all terms.

Overall, I am happy with the authors' response to feedback, in particular, their discussion of the data profile dimensions in "Detailed Analysis of the Five Solvers".

Below are a number of identified typos and small suggestions:

Line 32: it’s -> their
Line 39: We are motivated to notice -> We are motivated by the observation
Line 40: For example, data profiles -> For example, some data profilers

Line 45: Thus, the proposed multidimensional data profiles are more compact and effective in allocating a computational budget for different levels of accuracy.

Instead of 'Thus' perhaps say 'We argue that' or 'Numerical results indicate that'.

Line 75: NAa -> NMa

Line 77: with best -> with the best

Line 80: that is hybridizing -> that hybridizes

Line 81: Instead of using a restricted NM simplex to only one traditional design, the authors evolve NMa genetically to produce deterministic simplexes that can develop adaptive vertices through generations.

The sentence needs to be rewritten (simplified).

Line 86: and basically -> and that, basically, the

Line 93: upgrade the data profiles

Which data profiles?

Line 98: of trigonometric -> of the trigonometric

Line 111: When the next simplexes are characterized by different reflections, the HNMa performs not only similar reflections to that of the original simplex of the NMa, but also others with different orientations determined by the collection of non-isometric features. -> When different reflections characterize the next simplex, the HNMa performs similar reflections to that of the original simplex of the NMa and others with different orientations determined by the collection of non-isometric features.

Line 114: guaranteed to search the solution space of mathematical problems” -> guaranteed to search a higher proportion of the solution space

Line 118: calculated at the vertices -> calculated at vertices

Line 119: in ascending with -> in ascending order with

Line 120: $and$ -> and

Line 122: The need for Th arises when the HNMa performs a reflection in an axial component, it replaces the value of the axial component of the Th.

This sentence is not clear, please rewrite it.

Line 142: in extraction one -> in the extraction of one

Line 145 Therefore, the optimization solution of the HNMa not only reflects the opposite side of the simplex through the worse vertex, but also leads to implement various reflections that are determined by the collection of extracted features. -> Therefore, the optimization solution of the HNMa reflects the opposite side of the simplex through the worse vertex and leads to the implementation of reflections determined by the collection of extracted features.

Line 152: Solutions to the HNMa help extract the optimal characteristics of the non-isometric reflections in the reflected vertex and produce simplexes that lead to faster convergence rates than the original triangular simplex of the NMa.

Line 177 and 178: “ -> ``

Line 242: (in second), -> (in seconds)

Line 247: optimization classes -> optimization class.

Line 292: A simplex in some cases explores the neighborhood to update its threshold, but do not move only if the threshold is good enough to replace the worst point.

Grammatically, 'Do not' should probably be 'Does not'. However, the sentence makes more sense if 'but do not move' is changed to 'but moves'. Please rework this sentence.

Line 296: In this way, the HNMa mimics the amoeba style of maneuvering to move from one point to another when approaching a minimal point. -> In this way, the HNMa mimics an amoeba style of maneuvering from one point to another when approaching a minimal point.

Line 304: “ -> ``

Line 334: using 20 of 2.66 GHz -> using twenty 2.66 Ghz

Line 358: For example, MTNMa performs better to determine a solution to Quadratic (16) then Quadratic (8) in Table 1. -> For example, MTNMa performs better when optimizing Quadratic (16) as compared to Quadratic (8) in Table 1.


Line 359: While in other cases, the MTNMa generates fewer number of simplexes to approximate a particular solution for high dimensions than for less dimensions. -> While in other cases, the MTNMa generates fewer simplexes to approximate a particular solution for high dimensions than for lower dimensions.

Line 371: such environment.-> such an environment

Line 378: provide a complementary -> provide complementary

Line 396: requires the best possible reduction has to equal -> : requires that the best possible reduction has to equal

Line 407: By analyzing those five solvers of multi-directional trigonometric simplex designs, we have conducted further tests, which reveal that further evaluations are important to make decision on which solver should be used when there is a limited computational budget. -> We have conducted further tests by analyzing the five multi-directional trigonometric simplex solver designs. These reveal that higher dimensional data profiles are essential to deciding which solver should be used with a limited computational budget.

Line 410: solvers (1 and 5) already catch up, approximately after overtaking 400 simplex evaluations -> solvers (1 and 5) catch up after approximately 400 simplex evaluations

Line 417: This forms a strong argument that how a solver, in some cases, may always require a large number of simplex gradients but may have the potential to take less time to solve 100% of the test problems and improve the performance. -> This forms a strong argument as to how a solver, in some cases, may require a larger number of simplex gradients but may have the potential to take less time to solve 100% of the test problems.

Line 447: In the situation where Solver2 stands out as being the best of the five solvers, because certainly it requires not only less function evaluations but also less computational time than the other solvers. -> In this situation, Solver2 stands out as being the best of the five solvers because it requires fewer function evaluations and less computational time than the other solvers.

The paragraph starting on Line 451:

Regarding the statement on Line 452. While different mathematical problems are parallelizable to differing extents, I would argue that the significance of Z is more immediately dependant on whether the CF is implemented (or perhaps compiled) as a parallel function, whether the solver is implemented as a parallel algorithm and the effectiveness of the chosen parallelisation methods.

I suggest that this paragraph be simplified, for example:

The number of CPUs (Z) was not examined in our evaluation of the MTNMa and is included in formula (22) for completeness. This dimension is significant if an optimiser is deployed in a distributed environment such as Amazon Web Services (AWS). In such a case, the number of nodes in the virtual cluster is an important aspect of the computational budget and the inclusion of Z assists in the allocation of optimal numbers of CPUs for different solvers and for specific levels of accuracy.

Line 471: rotation property determined -> rotation determined

Line 471: To examine the performance of multiple solvers simultaneously, a linear model with a window of size 10 samples is suggested as the criteria that is used to make decision whether a solver is aborted or continued based on the direction vector of the window. -> When executing multiple solvers simultaneously, a linear model with a window of size 10 samples is suggested as the criteria by which a solver is aborted or continued based on the direction vector of the window.


Line 474: We also showed in this research that using the data profile as a function for allocating the computational budget and examining the relative performance of multiple solvers based only on the simplex gradients (one dimension) is not appropriate when the evaluation is expensive. -> We also showed in this research that using a data profile based only on the number of simplex gradients (one dimension) for allocation of the computational budget and examination of the relative performance of multiple solvers is not appropriate when cost-function (?) evaluation is expensive.

---

## Round 0.3 · accepted · Accept

The reviewer is satisfied with the revisions you made to this manuscript.

Reviewer 1 ·

Basic reporting

The authors have addressed all of my comments.

Experimental design

The authors have addressed all of my comments.

Validity of the findings

The authors have addressed all of my comments.

Additional comments

The authors have addressed all of my comments.